# Synthetic microbe communities provide internal reference standards for metagenome sequencing and analysis

Simon A. Hardwick[1,2], Wendy Y. Chen[1,2], Ted Wong[1], Bindu S. Kanakamedala[1], Ira W. Deveson[1,2], Sarah E. Ongley[3,4], Nadia S. Santini[5,6], Esteban Marcellin[7], Martin A. Smith [1,2], Lars K. Nielsen[7], Catherine E. Lovelock[8], Brett A. Neilan[3,4] & Tim R. Mercer[1,2,9]

The complexity of microbial communities, combined with technical biases in next-generation sequencing, pose a challenge to metagenomic analysis. Here, we develop a set of internal DNA standards, termed "sequins" (sequencing spike-ins), that together constitute a synthetic community of artificial microbial genomes. Sequins are added to environmental DNA samples prior to library preparation, and undergo concurrent sequencing with the accompanying sample. We validate the performance of sequins by comparison to mock microbial communities, and demonstrate their use in the analysis of real metagenome samples. We show how sequins can be used to measure fold change differences in the size and structure of accompanying microbial communities, and perform quantitative normalization between samples. We further illustrate how sequins can be used to benchmark and optimize new methods, including nanopore long-read sequencing technology. We provide metagenome sequins, along with associated data sets, protocols, and an accompanying software toolkit, as reference standards to aid in metagenomic studies.

[1] Garvan Institute of Medical Research, Sydney 2010 NSW, Australia. [2] St. Vincent's Clinical School, Faculty of Medicine, UNSW Sydney, Sydney 2052 NSW, Australia. [3] School of Biotechnology and Biomolecular Sciences, UNSW Sydney, Sydney 2052 NSW, Australia. [4] School of Environmental and Life Sciences, The University of Newcastle, Callaghan 2308 NSW, Australia. [5] Centre for Marine Bioinnovation UNSW Sydney, Sydney 2052 NSW, Australia. [6] Instituto de Ecologia, Universidad Nacional Autonoma de Mexico, Mexico City 04500, Mexico. [7] Australian Institute for Bioengineering and Nanotechnology, The University of Queensland, Brisbane 4072 Queensland, Australia. [8] School of Biological Sciences, The University of Queensland, Brisbane 4072 QLD, Australia. [9] Altius Institute for Biomedical Sciences, Seattle 98121 WA, USA. Correspondence and requests for materials should be addressed to T.R.M. (email: t.mercer@garvan.org.au)

The sequencing of DNA recovered directly from environmental samples can reveal the presence of microbial communities without requiring prior laboratory cultivation[1,2]. This approach, termed metagenomics, has exposed previously hidden microbial diversity in a range of different environments, from the open ocean[3], to complex soil samples[4], to the human microbiome[5]. Accordingly, metagenomics is often used to determine the profile of the microbes that inhabit a given environment, to diagnose the presence of a microbial pathogen[6] and to identify novel microbial lineages[7].

Comparisons between microbial communities that inhabit different environmental sites can also distinguish differences in the identity and abundance of microbes[8]. These approaches can identify microbes that confer specific environmental characteristics, or measure the impact of environmental variables on microbial communities, and have been used to discover host–microbe interactions, identify novel microbes with biotechnological value, and measure environmental health[9].

Despite the promise of this approach, the analysis of metagenomics data remains challenging. The sheer size and complexity of microbial genomes within a sample, many of which may be novel, confounds reliable identification and makes quantification of microbes difficult[1]. Additional technical biases that accrue during next-generation sequencing (NGS) further bias metagenomic analysis[2]. Differences in microbial population structures can also invalidate the assumptions that underlie normalization approaches, and thereby preclude accurate detection of genuine biological differences between samples[8,10].

Reference standards can offset these analytical challenges[11,12]. Reference standards enable the limits of sampling and analysis to be understood, and can measure technical variables that bias analysis with NGS. Reference standards can also evaluate quantitative accuracy, and act as scaling factors by which to normalize between samples. Accordingly, there is a pressing need to develop metagenome reference standards that can benchmark analytical methods and enable comparisons between multiple metagenome samples.

The National Institute of Standards & Technology (NIST) recently released a set of four bacterial reference genomes that can be used to benchmark and validate sequencing-based diagnostic assays[13]. Mock microbial communities—in which multiple microbes are individually cultured and combined at known abundances to form a community—are often favored as reference standards for metagenomics. Mock communities have proven useful for benchmarking different technologies, assessing biases, and for optimizing new analytical methods for metagenomics[5,14–18]. For example, the Human Microbiome Project assembled a mock community of bacteria and archaea commonly found on or in the human body[5,14]. More recently, groups have developed mock communities composed of microbes isolated from heterogeneous soil and aquatic environments[15,16]. However, a key limitation of mock communities is that they cannot be added directly to samples without the risk of contaminating downstream analysis. In contrast, synthetic spike-in controls can be added directly to samples to measure technical variation, and have been successfully used in human genome sequencing[19,20] and RNA sequencing[21,22].

Here, we have developed a set of 86 synthetic DNA standards termed "sequins" (sequencing spike-ins) that represent a synthetic microbial community. Sequins are formulated into a mixture emulating a synthetic microbial community that can be directly added to samples to act as qualitative and quantitative internal controls. We describe the design, synthesis, and validation of sequins, and show how they can be used to measure fold changes between microbial communities and facilitate intersample normalization. We provide metagenome sequins, along with associated data sets, protocols, and an accompanying software toolkit, as a resource to the research community at www.sequin.xyz.

## Results

**Design and validation of synthetic DNA standards.** We initially designed a set of 86 artificial DNA sequences (sequins) that represent the range, features, and complexity of a natural microbial community, despite having an entirely artificial sequence (see Supplementary Data 1 for details of sequences). We sampled a diverse selection of finished microbial genomes (RefSeq[23]) that encompassed a wide representation of taxa (including Eukaryota, Bacteria, and Archaea), size (0.5–10 Mb for prokaryotic genomes), GC content (20–71%), rRNA operon count (1–11), and isolation from a diverse range of environments (human body, aquatic, terrestrial, and extreme physical or chemical conditions) (Supplementary Fig. 1). A representative subsequence of each genome was selected and inverted to remove homology, while maintaining nucleotide composition, GC content, and the distribution of repetitive and unique sequences (Fig. 1a). Inverted sequences were queried against the BLAST non-redundant nucleotide collection (nt) database in order to ensure they had no significant homology (E value < 0.01) with any known natural sequences. To reduce the sequencing depth required to profile the community, we selected DNA subsequences from the microbial genomes that range in size from ~1 to 10 kb, resulting in a synthetic microbial community of total size ~227 kb.

We first simulated read libraries generated from metagenome sequins to validate synthetic sequences without the confounding impact of variables introduced through library preparation and sequencing. For comparison, we also simulated read libraries for a previously published mock community of 23 bacterial and 3 archaeal species (MBARC-26)[15,16]. Simulated libraries were aligned using Bowtie2[24] to the artificial sequences and MBARC-26 genomes.

The two libraries demonstrated comparable mappability (fraction of reads concordantly mapped) and no cross alignment of reads between sequins and MBARC-26 genomes was observed (Supplementary Fig. 2a). Simulating a range of matched sequencing depths (from 0.1 to 100× coverage) also indicated comparable breadth of alignment coverage for both sequins and MBARC-26 genomes across the full range (Fig. 1b).

We also performed de novo genome assembly of the simulated libraries (using Ray Meta[25]), showing that sequins and MBARC-26 genomes reached near-complete assembly (≥95% of reference genomes assembled into contigs) at ~20× mean coverage (Fig. 1c). Below this level, the assembly of sequins and MBARC-26 genomes were comparably affected by decreasing sequence coverage. We also found no evidence of chimeric assemblies between sequins and MBARC-26 genomes. Finally, a search of the simulated sequins library against BLAST's nt database using Centrifuge[26] returned no significant hits. Collectively, these simulated analyses provided sequence-level validation that sequins align and assemble equivalently to a wide range of microbial genomes, without risk of cross-alignments or chimeric assembly events contaminating downstream analysis.

**Synthesis and experimental validation of sequins.** Sequins were synthesized, manufactured, and formulated into a staggered mixture (Mix A) that spans a ~$3.2 \times 10^4$-fold concentration range (Supplementary Fig. 3a). We initially sequenced a neat mixture of sequins (i.e., with no natural DNA added) in triplicate to assess quantitative accuracy and technical variation between replicates. By plotting mean fold-coverage against known mixture

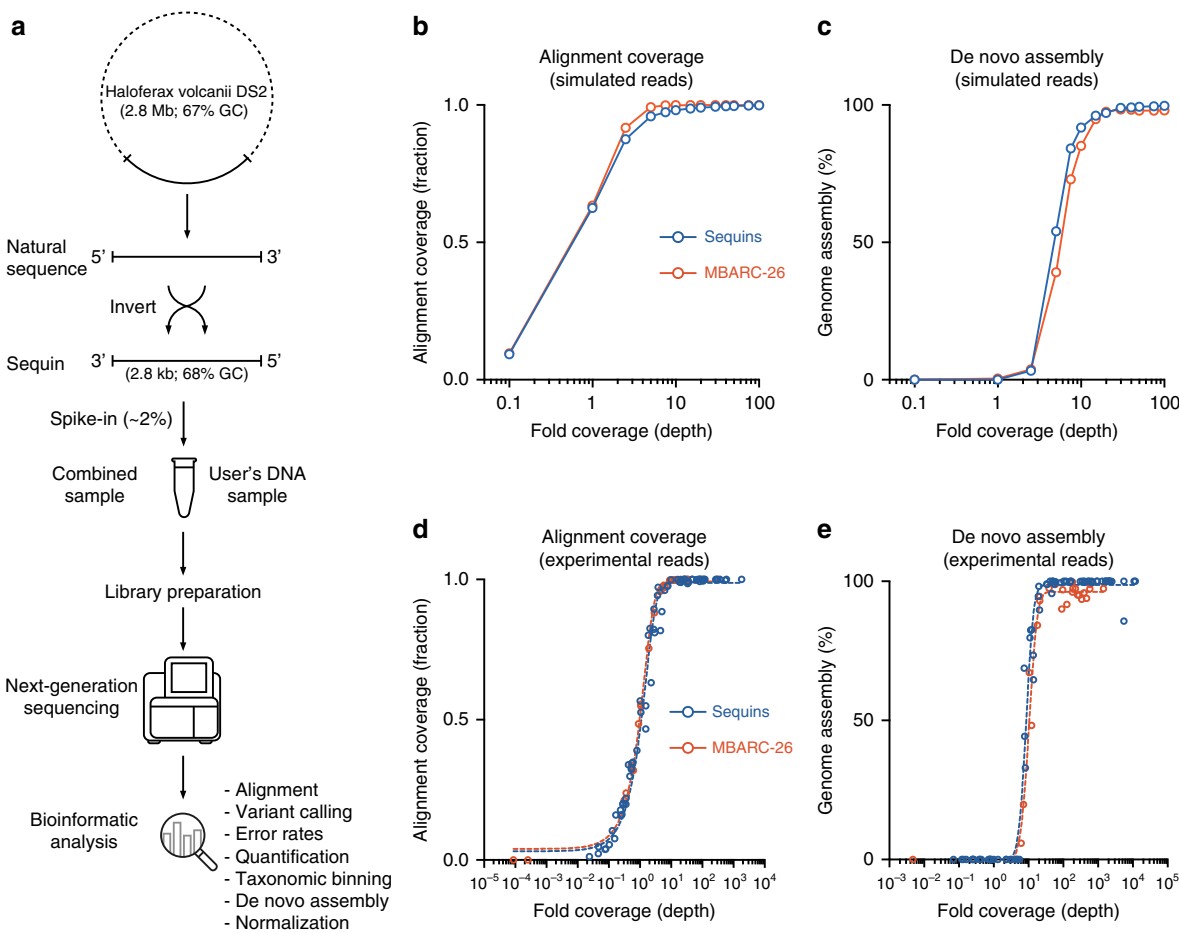

**Fig. 1** Schematic showing the design, use, and validation of DNA-sequencing spike-ins (sequins) for metagenomic analysis. **a** Metagenome sequins are designed by inverting a selected subsequence of a microbial genome that is then synthesized and combined with other DNA standards into a staggered mixture which represents a natural microbial community. Sequins are spiked into a user's DNA sample (at a low fractional concentration, e.g., 2%), undergoing combined library preparation, sequencing, and analysis. Examples of the bioinformatic steps that can be assessed using sequins are indicated. To validate the design of sequins, we generated simulated libraries that were not confounded by additional technical variables (**b**, **c**) and spiked sequins into a mock microbial community (MBARC-26) and sequenced the combined sample (**d**, **e**). Sequins (blue) displayed comparable breadth of alignment coverage to real microbial genomes (red) across the full range of fold-coverage depths observed in both simulated (**b**) and experimental (**d**) data. We also observed equivalent de novo assembly of sequins and MBARC-26 genomes at matched fold-coverage levels in both simulated (**c**) and experimental (**e**) data. The dashed lines in **d** and **e** were fitted using Richard's five-parameter dose–response curve

concentration, we observed high quantitative accuracy ($R^2 = 0.979$; slope $= 1.04 \pm 0.02$) and minimal variation between replicates (Fig. 2a), with reproducible sequencing coverage profiles for each sequin (Fig. 2b).

We next aligned the neat sequins library to a combined index comprising sequins and MBARC-26 genomes, finding that only a negligible fraction of the reads (0.26%) aligned to an MBARC-26 genome (Supplementary Fig. 2b). The vast majority (>99%) of these cross-aligning reads aligned to the *E. coli* K-12 genome (NC_000913.3), and likely result from contamination in laboratory reagents and processes[27,28]. Conversely, no experimental reads derived from a neat preparation of MBARC-26 gDNA (SRR3656745) aligned to any sequin genome (Supplementary Fig. 2b). The two libraries demonstrated comparable mappability, with concordant pair alignment rates of 89.7% and 86.9% for sequins and MBARC-26 genomes, respectively (Supplementary Fig. 2b).

To enable assessment of fold change differences between samples, we also prepared an alternative mixture of sequins (Mix B) containing the same set of 86 DNA standards, but with a subset that undergo known fold changes between mixtures ($n = 50$) and a subset that remain at equimolar concentrations ($n = $

36) (Supplementary Fig. 3b). This alternative mixture design allows the accuracy of fold changes across different samples to be assessed, while also providing negative controls for inter-sample normalization[29].

**Validation of sequins using MBARC-26 mock community**. To validate the use of metagenome sequins, we next spiked the staggered mixture into genome DNA from the MBARC-26 mock community at a low fractional abundance (1%). The combined sample then underwent concurrent library preparation and sequencing (Fig. 1a; see Methods). Of the resulting library, 1.49% of reads aligned to sequins, while 98.1% aligned to MBARC-26 genomes. We plotted the breadth of alignment coverage for each sequin and MBARC-26 genome against its measured fold-coverage, showing that they performed comparably across the full range (Fig. 1d). The rate of errors in these sequenced reads was comparable for sequins and MBARC-26 genomes, confirming the use of sequins for estimating run-specific sequencing error rates (Supplementary Fig. 2c).

Since sequins span the full range of GC contents observed in natural microbial genomes (~24–72%), we used sequins to assess the extent of any GC bias in the sequencing data. We plotted the

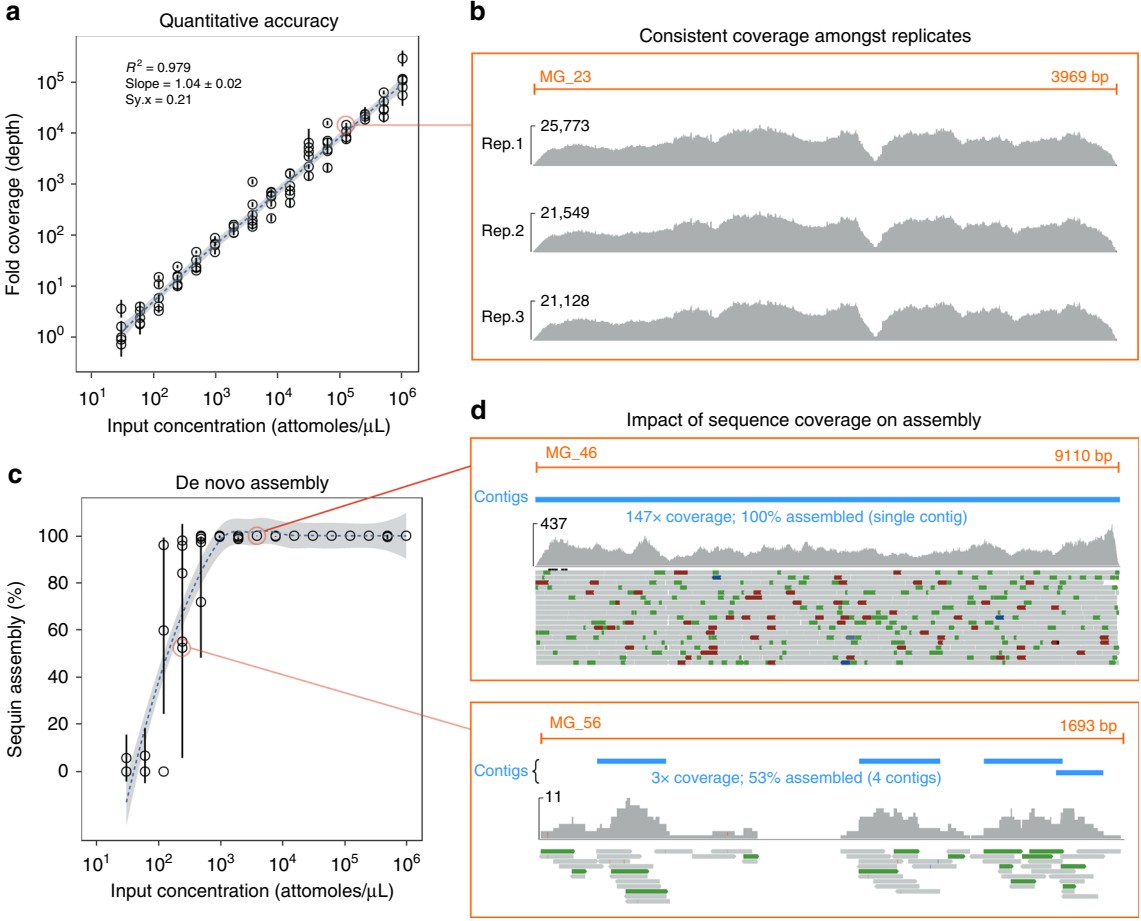

**Fig. 2** Using sequins to assess quantitative accuracy and de novo assembly. **a** Scatter plot shows measured fold-coverage for each sequin plotted against its known input concentration, in triplicate. Error bars indicate standard deviation (SD) between replicates. A linear regression model is fitted to the data (dotted line) with 95% confidence interval shown (gray shading). **b** By sequencing a neat preparation of the staggered sequins mixture (Mix A) in triplicate, we observed the same coverage patterns and biases in each replicate. Example shows genome browser views of the sequencing coverage for three replicates of a randomly chosen sequin, *MG_23*. **c** Scatter plot shows the fraction of each sequin de novo-assembled plots against its known input concentration, in triplicate. Error bars indicate SD between replicates. **d** Genome browser views of two sequins demonstrate the impact of sequence coverage on assembly. *MG_46* (top) shows high sequence coverage, and was fully assembled in a single contig. By contrast, *MG_56* (bottom) has low sequence coverage and was only partially assembled with four fragmented contigs

measured sequencing error rate for each individual sequin against its respective GC content, finding that the mismatch error rate increased steadily with increasing GC content (Supplementary Fig. 4a, left). This effect was also observed for indel errors, albeit significantly weaker. We observed similar results for the MBARC-26 genomes (Supplementary Fig. 4a, right). We also plotted fold-coverage (observed divided by expected) for each sequin against its respective GC content, observing only very minor GC bias, with slightly elevated coverage for sequins in the middle of the GC content range (Supplementary Fig. 4b, left). Again, similar results were seen for MBARC-26 genomes (Supplementary Fig. 4b, right). While previous studies have found much stronger under-representation of GC-rich and AT-rich fragments[30,31], all libraries in the present study were prepared using a PCR-free protocol (see Methods) and thus the effect of any GC bias would be expected to be less obvious. To investigate any length bias, we plotted observed/expected coverage of each sequin against its length, finding slightly reduced coverage at both the small and large ends of the spectrum (Supplementary Fig. 4c).

We next assessed the de novo assembly of the sequins by comparison to the MBARC-26 genomes. We found that, despite their reduced size, sequins assembled similarly to MBARC-26 genomes at matched coverage levels (Fig. 1e). Both sequins and

MBARC-26 genomes achieved complete assembly at ~30× coverage, a finding consistent with previous studies of bacterial genomes[32]. Again, no chimeric contigs between sequins and MBARC-26 genomes were observed in the assembly. The comparable performance of sequins and MBARC-26 genomes demonstrated the commutability of sequins as an internal reference for addition to environmental DNA samples during metagenome sequencing and analysis.

**Benchmarking of long-read sequencing using sequins**. Long-read sequencing can resolve repetitive regions within a microbial genome, and is useful for genome assembly[33]. Therefore, we next used the sequins to benchmark and optimize the performance of Oxford Nanopore Technology's MinION long-read sequencing technology. We first sequenced a neat mixture of metagenome sequins, generating a total of 160,709 base-called reads, of which 74.0% successfully aligned to reference sequences using bwa-mem (Supplementary Table 1). We detected 75 (out of 86) sequins with at least one read, corresponding to a limit of detection (LoD) of 60 attomoles/µL and a dynamic range of $1.6 \times 10^4$. Most sequins (61 out of 86) were fully covered by at least one MinION read, thereby obviating the need for de novo assembly (Supplementary

Fig. 5a). To assess quantitative accuracy, we plotted the measured fold-coverage of each sequin against its input concentration, noting that the quantitative accuracy of MinION sequencing ($R^2$ = 0.914) was slightly less than matched Illumina short-read data ($R^2$ = 0.981) (Supplementary Fig. 5b).

We next measured sequencing error rates, finding that MinION had a mismatch error rate of 7.12% (compared to 0.127% for Illumina) and an indel rate of 8.71% (compared to 0.00770% for Illumina). Notably, MinION sequencing suffered significantly higher indel rates in homopolymeric sequences (mean indel rate = 16.7%) compared to other regions (mean = 7.69%; unpaired $t$-test with Welch's correction, $p$ value <0.0001), with characteristic sequence coverage drops at the upstream end of homopolymer tracts (Supplementary Fig. 5c). This phenomenon has also been reported by others[33–35].

We then spiked MBARC-26 mock community DNA with metagenome sequins (5% fractional abundance) and sequenced the combined library to generate 299,050 base-called reads, with a mean length of 2590 bp (Supplementary Fig. 6a). While all MBARC-26 genomes were detected with at least one read, the average fold-coverage varied from as low as 0.0439× for *Nocardiopsis dassonvillei* up to 24.4× for *Fervidobacterium pennivorans* (Supplementary Fig. 6b). We also detected 66 out of 86 sequins, corresponding to an LoD of 120 attomoles/µL and a dynamic range of $8.2 \times 10^3$. The error rates for sequins and MBARC-26 genomes were comparable, with mismatch rates of 8.30% and 7.67%, respectively, and indel rates of 6.64% and 6.49%, respectively (Supplementary Fig. 6c).

We then used the sequins to benchmark the performance of a range of different read-mapping tools designed for long, error-prone reads (bwa-mem, graphMap, marginAlign, and minimap2; Supplementary Table 2). Across the four mappers, the fraction of reads aligning to sequins ranged from 7.21 to 8.71%, with a corresponding range in base-level sensitivity (i.e., the fraction of reference bases covered by aligned reads) from 68.9 to 73.2% (Supplementary Fig. 6d). We also used sequins to assess quantitative accuracy, finding that average fold-coverage for sequins ranged from 97.3 to 111.7×, and quantitative accuracy ($R^2$) ranged from 0.889 to 0.915. This illustrates how sequins can provide a useful internal metric by which to benchmark and optimize bioinformatic analysis.

**Using sequins in a real metagenome experimental context.** Having validated the veracity of sequins, we next sought to demonstrate their use in a real experimental context. We spiked sequins into DNA extracted from saltmarsh samples collected from Haslam's Creek, in Sydney Olympic Park, Australia (see Methods). We sought to compare the microbial composition of (i) non-rooted soil, (ii) rhizosphere, and (iii) root samples taken from saltmarsh sites undisturbed by human development (termed "natural"), with saltmarsh sites that were regenerated as part of the Sydney 2000 Olympics (regenerated) (Supplementary Fig. 7a). Therefore, we spiked Mixes A and B alternately into DNA extracted from three replicate natural and regenerated sites, respectively, at 5% fractional abundance prior to library preparation and sequencing.

To assess the quantitative accuracy of fold change measurements between samples, we plotted observed against expected $\log_2$ fold change (LFC) for each sequin, indicating a strong linear relationship ($R^2$ = 0.971; slope = 1.05 ± 0.02) (Fig. 3a). We then performed differential abundance testing on the sequins (Mix A vs. B) using DESeq2[36,37]. Of the sequins that change abundance across mixtures, 48 out of 50 returned an adjusted $p$ value <0.05 (sensitivity = 96.0%). Conversely, of the negative control sequins, 32 out of 36 returned a $p$ value (adj) >0.05 (specificity = 88.9%).

As expected, diagnostic power increased with expected LFC, as illustrated by receiver operator characteristic (ROC) analysis (Supplementary Fig. 7b).

De novo assembly of metagenome samples can be confounded by both inherent sample complexity, as well as technical factors introduced during library preparation, sequencing, or analysis. Sequins can distinguish between these outcomes and thereby assist in quality control and troubleshooting. To demonstrate this, we performed de novo assembly on all samples using Ray Meta, and evaluated the quality of each assembly using MetaQUAST (supplying the sequin reference sequences; Supplementary Table 3). The N50 values for contigs aligned to sequins (2272–2879 nt) indicated that library preparation, sequencing, and bioinformatic assembly were performed successfully (Supplementary Fig. 7c). Conversely, the N50 values for unaligned contigs were poorer and less consistent (697–3659 nt), indicating that inherent sample complexity limited assembly, rather than technical factors.

We next examined the taxonomic composition of reads using MG-RAST[38], by searching against the RefSeq[23] database (phylum level). Principal component analysis (PCA) revealed that samples clustered loosely by type (non-rooted soil, rhizosphere, and root) rather than environment (natural and regenerated) (Fig. 3c, left). However, box plots of relative log expression (RLE)[39] showed a clear need for normalization, with large distributional differences between samples that indicated unwanted variation (Fig. 3c, right). While RLE plots were first developed for gene expression data, they can additionally be used to uncover unwanted variation in many other types of high-dimensional data, and are helpful for determining whether a normalization procedure has worked as intended[39].

To normalize between samples, we employed the RUVg method, which performs factor analysis on suitable sets of control genes (e.g., spike-in controls) to adjust for unwanted technical variation[40]. To perform RUVg normalization, we nominated the subset of sequins that remain at equimolar concentrations across mixtures as negative controls ($n$ = 36). RUVg normalization improved the data, with samples still clustering loosely by type (Fig. 3d, left), and sample RLE plots now centered around zero with most of the excessive variation removed (Fig. 3d, right). While samples still did not cluster perfectly by type, it must be kept in mind that the replicates were collected from three separate natural and regenerated sites. Nonetheless, the fact that samples clustered loosely by type but not at all by environment is an interesting biological observation, and implies that regeneration of these saltmarsh sites after initial development did not have a discernible effect on the microbial ecology of the samples. In order to compare variation among replicates, we plotted the mean coefficient of variation for each of the six treatment groups both before and after RUVg normalization, finding that variation decreased substantially after RUVg normalization (Fig. 3b). This provided further evidence that RUVg normalization was successful. Notably, the use of sequins with the RUVg approach outperformed other normalization methods, including upper-quartile (UQ) normalization, where some samples still displayed excessive variability (see Supplementary Figs. 8 and 9).

**Using sequins for normalization between samples.** While metagenomics can profile the relative proportions of species (or other taxonomic units) within a sample, it is generally not possible to detect differences in total microbial load between samples[8]. This limitation can be addressed by the addition of spike-in controls in fixed (rather than fractional) amounts to provide a reference by which to rescale and normalize samples[10,41].

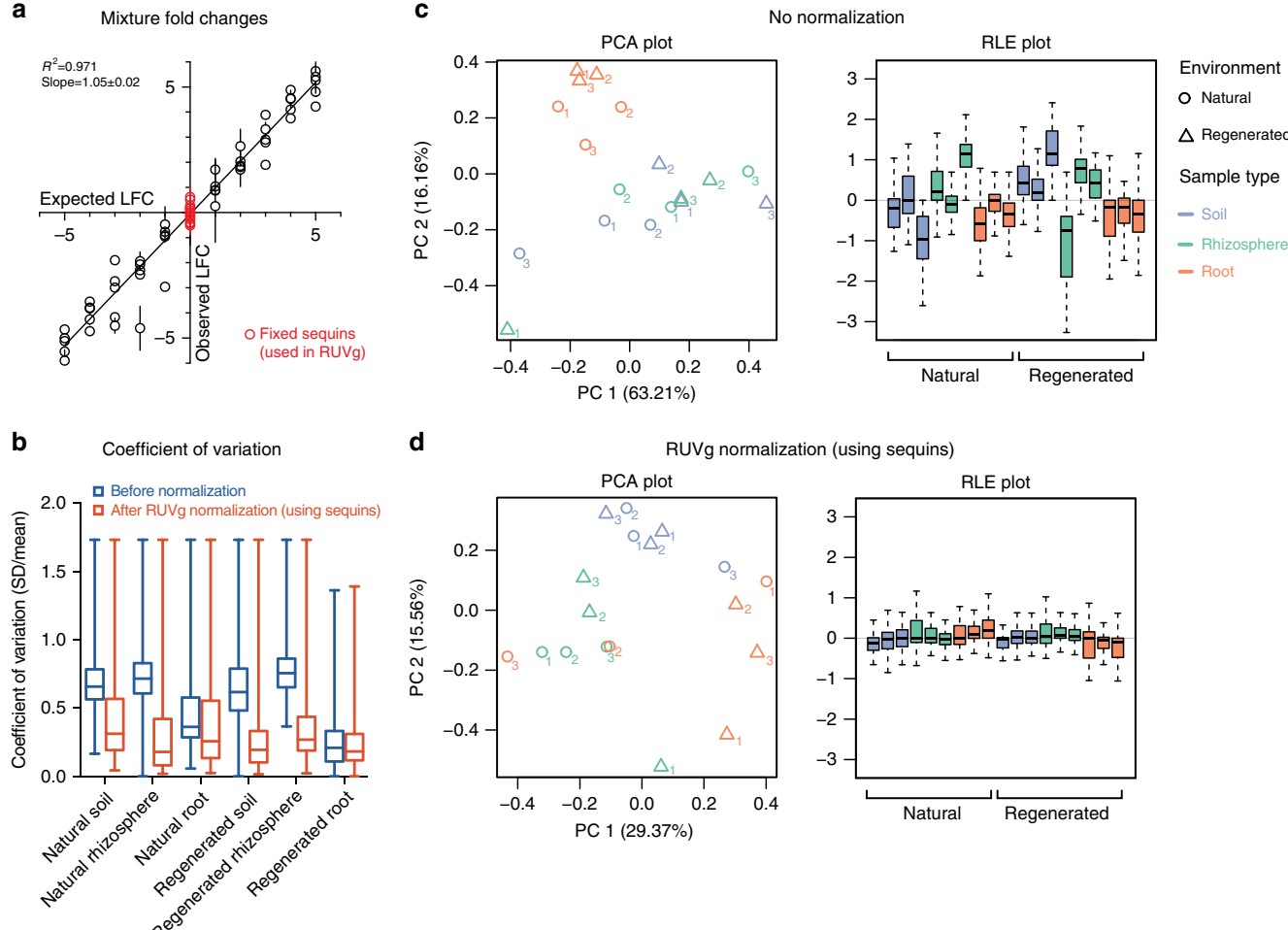

**Fig. 3** Using sequins to assess fold changes and normalize between real metagenome samples of unknown content. **a** Scatter plot shows observed $\log_2$ fold change (LFC) against expected LFC for each sequin across Mixes A and B (nine replicates of each). Error bars indicate standard deviation (SD) among replicates. The subset of sequins that remain fixed between mixtures (red) are used in RUVg normalization. **b** Box plots show the coefficient of variation (SD/mean) between replicates for each phylum within each sample type, both before (blue) and after (red) RUVg normalization. Box center lines indicate median; bounds of boxes indicate upper and lower quartiles; whiskers extend to min/max values. **c** Without normalization, samples clustered loosely by sample type rather than environment, as shown by principal component analysis (PCA; left). However, relative log expression (RLE) box plots demonstrated a clear need for normalization, with samples not centered around zero and wide differences in variation between samples (right). **d** After performing RUVg normalization using sequins, samples still clustered by sample type (left), and RLE plots showed a marked improvement with samples now centered around zero and the excessive variation of most samples removed (right). Quantification results are based on phylum-level abundance estimates from read mapping (using MG-RAST; $n = 75$ phyla)

To demonstrate this principle, we assembled two distinct mock microbial mixtures (A and B) comprising gDNA extracted from four species of Cyanobacteria (Fig. 4a), with the total amount of genome DNA doubling each time (A1, A2, and A3; and B1, B2, and B3). A fixed (as opposed to fractional) amount of sequins was added to each sample before library preparation and sequencing. Comparison of these mixtures provided examples of both relative and absolute fold change comparisons for each microbe.

We initially performed conventional normalization (based on genome size and sequencing depth). While this enabled the relative abundance of a microbe fraction to be measured within a single mixture, this approach was unable to distinguish absolute changes in abundance. Accordingly, the microbial composition of Mixes A1, A2, and A3 were indistinguishable from one another; and likewise for Mixes B1, B2, and B3 (Fig. 4b). By contrast, after normalizing each sample relative to the sequin input, this allowed the increasing absolute abundance of microbes to be detected within the mixtures (Fig. 4c).

The advantage of normalizing by absolute comparison to sequins was further demonstrated by assessing microbe fold change differences between mixtures. For example, if we compare the abundance of the *Synechocystis sp. PCC 6803* genome between the six mixtures, we observe a total of 15-fold change differences (Fig. 4d). We are unable to accurately resolve either relative or absolute fold change differences when normalizing our sample by genome size and library depth. By contrast, the fold changes in *Synechocystis* abundance could be accurately resolved when normalized relative to absolute sequins.

To demonstrate the breadth of this advantage, we plotted the observed relative to expected fold change for each species across the six samples (total of 60 comparisons), observing a much stronger correlation following normalization with sequins ($R^2 = 0.997$), compared to normalizing only by genome size and library depth ($R^2 = 0.537$) (Fig. 4e). This analysis illustrates how sequins can be used to reveal major global shifts in microbe community abundance that are otherwise imperceptible using conventional normalization procedures.

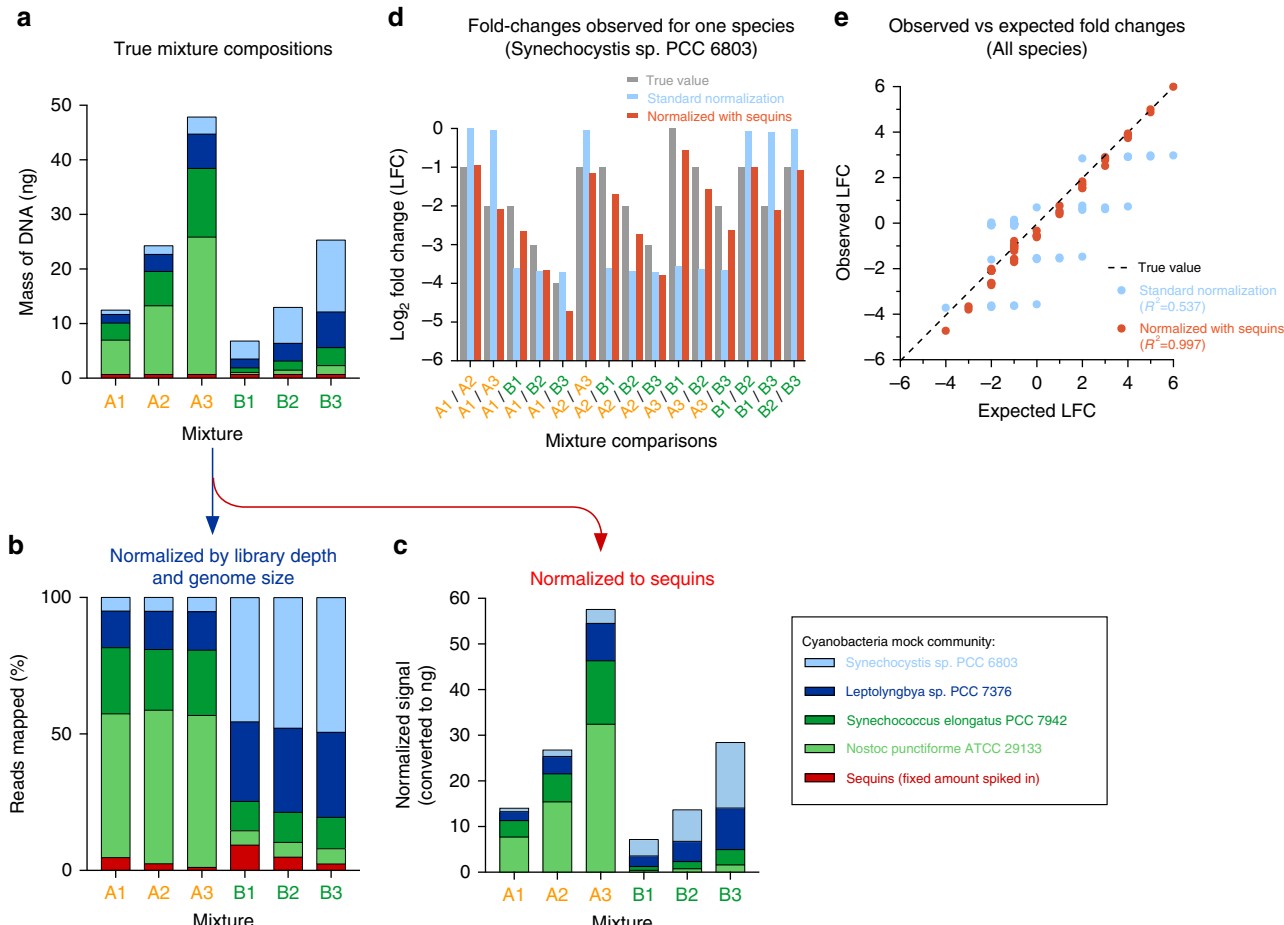

**Fig. 4** Using sequins to accurately resolve both relative and absolute changes in microbial load. **a** We assembled six mock microbial mixtures comprising different amounts of four distinct cyanobacteria species. A fixed (as opposed to fractional) amount of sequins (red) was added to each mixture. **b** Normalizing on the basis of library depth and genome size enabled the fractional composition of each mixture to be accurately measured; however, Mixes A1, A2, and A3 were indistinguishable from each other (likewise for Mixes B1, B2, and B3). However, as the amount of microbial DNA increased, the fraction of reads aligning to sequins decreased concomitantly. **c** By dividing the fractional abundance of each species by the fraction of reads aligned to sequins in each mixture, this enabled each sample's reference point to be rescaled, thereby allowing the accurate resolution of both relative and absolute abundance. **d** We tracked the fold changes observed for the *Synechocystis sp. PCC 6803* genome across the six mixtures, providing 15 separate fold change comparisons, including both relative and absolute abundance shifts. Bar graph shows the log2 fold changes (LFC) observed after performing normalization on the basis of genome size and library depth (blue), and after carrying out additional normalization with sequins (red). The true LFC value is shown in gray. **e** Scatter plot shows the measured LFC for each species plotted against the expected LFC (a total of 60 comparisons). $R^2$ value indicates the correlation coefficient obtained after carrying out linear regression analysis

## Discussion

Metagenomics can profile the community of microbes within an environmental sample. This analysis of microbes does not require prior laboratory cultivation, can discover new microbial lineages[7], and diagnose the presence of pathogens within patient samples[6]. Despite this promise, the analysis and comparison of metagenome samples is challenging, and reference standards are needed to ensure the accuracy and reproducibility of results[12].

Here, we describe the development of reference standards, termed "sequins," for metagenome experiments. Sequins comprise a set of synthetic DNA standards that mimic the sequence complexity, phylogenetic composition, and GC content of a natural microbial community, yet have no homology to known nucleic acid sequences. This enables their use as internal reference standards for downstream steps, including library preparation, sequencing, and bioinformatic analysis.

Metagenome samples can be affected by a range of unwanted technical variables, such as different library preparation methods[42] or the presence of enzymatic inhibitors[43]. A major advantage of sequins is their ability to measure and mitigate this technical variation that influences sequencing. As internal standards, sequins can be used to normalize different methods, experiments, and batches, and distinguish genuine biological differences from confounding technical variables[40].

In addition to enabling quality control and inter-sample normalization, reference standards are essential for the development and optimization of new sequencing technologies[12,15]. In this study, we used sequins to benchmark the performance of both short-read and long-read sequencing technology, as well as a range of different software tools. Emerging long-read technologies offer several advantages for metagenomics, including more accurate resolution of repeat elements[33], and the ability to carry out real-time, portable genomic surveillance in the field during disease outbreaks[44]. As new technologies continue to develop, sequins provide a constant reference by which to benchmark different tools.

Global differences in the size and complexity of microbial populations can violate the assumptions that underlie

                                                          

normalization between samples[10]. For example, the comparison of the taxonomic subpopulations within a microbial community is limited to the relative abundance of taxa, and metagenomics cannot detect shifts in absolute abundance between samples[8]. Here, we demonstrate how sequins can be used to calibrate and rescale different samples, thereby enabling the accurate measurement of both relative and absolute fold changes.

Metagenome studies often sample widely disparate environmental sites, and often at different times and methods, resulting in substantial batch variation between experimental samples. We propose that the routine use of reference standards, such as sequins, provides a reliable reference against which to standardize sample collection across such different sites, and can support more rigorous meta-analysis between different experiments and studies.

Within the laboratory, sequins can be used for the routine surveillance of individual samples, and to measure and improve operational performance. This is particularly important in clinical microbiology, where equipment and processes must be regularly validated, and the diagnosis of pathogens within patient samples assured[6,13]. Indeed, we expect that reference standards such as sequins will be useful for clinical assay development and validation, and also for ongoing quality control and proficiency testing procedures[6,45].

Given these benefits to metagenome methods, we have made sequins, along with associated data sets, protocols, and an accompanying software toolkit, freely available to researchers on our website at www.sequin.xyz.

## Methods

**Design of artificial microbial community**. To design artificial sequences, we first retrieved all high-quality, finished, microbial genome sequences from RefSeq[23]. We then ranked and systematically selected genomes according to a range of features, including taxa (including representatives from Eukaryota, Bacteria, and Archaea), size (0.5–10 Mb for prokaryotes), GC content (20–71%), rRNA operon count (1–11), and isolation from a diverse range of environments (human body, aquatic, terrestrial, and extreme physical or chemical conditions) (Supplementary Fig. 1). This ensured that the selected genomes provided a proportional representation of these different features. To illustrate the evolutionary relationships between these microbial genomes, we generated a phylogenetic tree using iTOL (v3) with default parameters[46].

We then aimed to select a subsequence representative of each genome as follows. We first selected a large number (>100) of random ~1–10 kb subsequences of each genome. Subsequences were then inverted to remove primary sequence homology, while still retaining different features of the original microbial genome, e.g., size, nucleotide composition, GC content, and the distribution of repetitive and unique sequences. Inverted sequences were then queried against the BLAST nt database for homology to known sequences and any inverted sequence with a significant match ($E$ value < 0.01) was omitted from further analysis. From the remaining subsequences, we then selected a subsequence that had a GC content closest to the original source genome. In addition, some sequences were manually curated to generate sequins with extremely high or low GC content. Upon completion, we generated a set of 86 synthetic DNA standards (sequins), which ranged in length from ~1 to 10 kb, with a combined size of ~227 kb (see Supplementary Data 1 for full details of sequences).

**Simulated read libraries**. To validate that sequins perform equivalently to real microbial genomes, we compared them to a mock microbial community consisting of 26 bacterial and archaeal species (MBARC-26)[15,16]. We generated simulated read libraries for sequins and MBARC-26 genomes using the ART simulator[47] (vMountRainier-2016–06–05) in Illumina HiSeq mode with 2 × 125 bp reads. The FASTA sequences inputted to ART were derived from the reported NCBI Reference Sequence Database accession numbers for MBARC-26 genomes[16]. We simulated libraries corresponding to a range of sequencing depths, from 0.1 to 100× coverage.

**Synthesis and manufacture of sequins**. DNA standards were synthesized by GeneArt (Life Technologies) and cloned into pMA vectors. A list of all sequences, as well as their concentration in each mixture, is provided in Supplementary Data 1. The synthesized plasmid each sequin was transformed in E. coli (α-select Silver Efficiency, Bioline, Australia), grown up in a 50 mL culture, purified and used for DNA sequence verification by Sanger sequencing. Sequins were excised from the plasmid backbone by restriction digest and subsequent

confirmation with gel electrophoresis. Purified sequins were quantified using the BR dsDNA Qubit Assay on a Qubit 2.0 Fluorometer (Life Technologies) and verified on the Agilent 2100 Bioanalyzer with an Agilent High Sensitivity DNA Kit (Agilent Technologies).

**Preparation of sequins mixtures**. Staggered mixtures (A and B) each consist of a pool of 86 sequins that are combined at twofold serial dilutions to encompass ~$3.2 × 10^4$-fold concentration range (Supplementary Fig. 3a). Each mixture has 16 staggered concentration points, with at least 5 sequins (representing a range of GC contents and lengths) per point. Individual sequins in the two alternative mixtures are present at defined molar concentration ratios, allowing for comparison of fold changes in abundances between samples (Supplementary Fig. 3b). The 86 sequins were pooled using an epMotion 5070 epBlue™ software program to make the final mixtures.

**Microbe and metagenome samples**. Genomic DNA from the MBARC-26 mock community[16] was obtained from the laboratory of T. Woyke (Joint Genome Institute, CA, USA).

Saltmarsh samples were collected from natural and regenerated sites along Haslam's Creek within Sydney Olympic Park, Australia (33°50′24.04″S, 151°3′48.57″E and 33°50′24.62″S, 151°3′31.69″E, respectively) in February 2016. We collected triplicates of non-rooted soil, rhizosphere, and root material from the first 10 cm of topsoil at three natural sites (natural 1, natural 2, and natural 3) and three regenerated sites (regenerated 1, regenerated 2, and regenerated 3). All materials were transported from the field to the laboratory within vials in a dry shipper and stored at −86 °C until further analyses were performed. In brief, we term non-rooted soil as root-free sediment, rhizosphere soil samples were obtained by washing up roots with a standard phosphate-buffered saline solution (PBS) and once washed in PBS, root material was also collected in a different vial. DNA extractions were carried out using an Mo Bio PowerSoil® DNA Isolation Kit according to the manufacturer's instructions.

For the cyanobacteria mixtures, we used gDNA extracted from monocultures of four different species for which complete, finished genome references are available: Nostoc punctiforme ATCC 29133 (NC_010628.1), Synechocystis sp. PCC 6803 (NC_000911.1), Synechococcus elongatus PCC 7942 (NC_007604.1), and Leptolyngbya sp. PCC 7376 (NC_019683.1). We prepared two alternative mixtures of gDNA (A and B), with each species undergoing a known fold change between mixtures (Fig. 4a). Mixture A was sequenced three times, with the amount of DNA added doubling each time (A1, A2, and A3); likewise for Mix B (B1, B2, and B3). A fixed amount of sequins (0.666 ng) was added to each sample prior to library preparation and sequencing. To normalize samples using sequins, we calculated the fractional abundance of each species (after first normalizing for genome size and library depth) and then divided this by the fraction of reads, which aligned to sequins in each sample.

**Preparation of Illumina DNA libraries and sequencing**. Initially, metagenome sequins (Mixes A and B) were sequenced neat (i.e., without any natural DNA added). Mix A was sequenced in triplicate, in order to assess technical variation. For the MBARC-26 validation experiment, sequins were spiked into microbial gDNA at 1% fractional abundance. For the saltmarsh samples, sequins were spiked into total metagenome DNA at 5% fractional abundance. The Nextera XT Sample Prep Kit (Illumina®) was used to prepare DNA libraries according to the manufacturer's instructions. Prepared libraries were quantified on a Qubit Fluorometer (Life Technologies) and verified on an Agilent 2100 Bioanalyzer with an Agilent High Sensitivity DNA Kit (Agilent Technologies). Libraries were sequenced on a HiSeq 2500 instrument (Illumina®) with 2 × 125 bp reads at the Kinghorn Centre for Clinical Genomics, Sydney, Australia.

**Read alignment and de novo assembly**. FastQC (v0.11.5) (http://www.bioinformatics.babraham.ac.uk/projects/fastqc/) was used to confirm sequence quality of reads for the experimental data sets. Reads were trimmed using Cutadapt (v1.8.1) with the Trim Galore wrapper (v0.4.1) in --paired mode. Simulated and experimental reads (FASTQ files) were mapped with Bowtie2[24] (v2.3.2) to a combined genome comprising sequin DNA sequences and microbial genomes. Average fold-coverage of each genome was calculated from mapped BAM files using the BEDTools[48] "genomecov" feature (v2.25.0). To assess GC content bias, we calculated observed/expected coverage as the mean coverage of each sequin divided by its expected concentration in the mixture. This controls for the fact that sequin abundances vary over several orders of magnitude within the staggered mixtures. Breadth of alignment coverage was calculated using the BEDTools "coverage" feature. De novo genome assembly was carried out using Ray Meta[25] (v2.3.1) with default parameters. Assemblies were evaluated using MetaQUAST[49] (v4.5) by supplying reference genome sequences. In order to test whether reads derived from real microbial genomes align to any sequins genome, we used a previously published data set, in which MBARC-26 gDNA was sequenced neat[16] (~174 million paired-end 150 bp reads sequenced on Illumina HiSeq 2000; NCBI Accession: SRR3656745). Illumina sequencing error rates (mismatches and indels) were calculated using the "AlignmentSummaryMetrics" tool from the Picard suite

(http://broadinstitute.github.io/picard). IGV[50] (v2.3.57) was used to visualize all read alignments and assembled contigs.

**Nanopore long-read sequencing**. We sequenced a neat preparation (Mix A) of metagenome sequins (1 µg) on a single flow cell using the MinION platform (Oxford Nanopore Technologies). We also sequenced MBARC-26 gDNA (700 ng) on a separate flow cell after spiking with sequins (Mix A) at 5% fractional abundance. Libraries were prepared in accordance with the manufacturer's instructions and sequencing was carried out using LSK108 chemistry. Base calling was performed using the MinKNOW software (v1.7.3). We used poretools[51] (v0.6.0) to process the raw FAST5 files, and aligned them to reference sequences using four different mappers designed for long error-prone reads: bwa-mem[52] (v0.7.12) with the –x ont2d option; GraphMap[34] (v0.5.2); marginAlign[35] (v0.1); and the new minimap2 tool (v2.0-r191-dirty) (https://github.com/lh3/minimap2). The latter three tools were run with default parameters. Nanopore sequencing error rates (mismatches and indels) were calculated using the "AlignmentSummaryMetrics" tool from Picard. To calculate indel rates in homopolymeric sequences, we first retrieved the insertion and deletion rates for each nucleotide in the sequins reference sequences using the piledriver tool (https://github.com/arq5x/piledriver). We then annotated the sequins references for homopolymers using the python script findSimpleRegions_quad.py, specifying a minimum homopolymer length of 6 bp (available from https://github.com/ga4gh/benchmarking-tools/tree/master/resources/stratification-bed-files/LowComplexity). Finally, we calculated the average per-base indel rate for homopolymers, and compared it with the rate for the remainder of the reference sequences.

**Taxonomic binning and differential abundance analysis**. The MG-RAST pipeline[38] (v4.0.2) was used to examine the taxonomic composition of each saltmarsh sample, by inputting raw sequence reads directly. Default options for quality control, including removal of artificial duplicate reads, trimming, and screening for human DNA, were used. Reads were compared against the RefSeq[23] database (domain and phylum levels) using BLAT parameters of 60% similarity, 15 bp and $E$ value of $10^{-5}$. Receiver operator characteristic (ROC) curves were constructed by modifying the code from the ercc-dashboard R package[29]. Samples were normalized using the RUVg procedure, using the subset of 36 sequins that remain at equimolar concentrations between mixtures as negative controls, with $k = 1$ factors of unwanted variation[40]. We used DESeq2[36] to test for differential abundance between samples at both the domain and phylum levels, including fold changes of sequins between Mixes A and B.

**Software package**. We have developed a software package (termed "ANAQUIN") to assist users in the analysis of data containing sequins (or other spike-in controls) [53]. The software is available for download as a Bioconductor package at https://bioconductor.org/packages/release/bioc/html/Anaquin.html.

**Data availability**. DNA-sequencing libraries are available from SRA under BioProject ID PRJNA422917. Saltmarsh metagenome samples have been uploaded to MG-RAST (project number MGP81258). In addition, selected data sets are available online at our website, www.sequin.xyz.

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

## Acknowledgements

The authors thank Dr T. Woyke (Joint Genome Institute, CA, USA) for providing MBARC-26 mock community DNA. The authors also thank the following funding sources: Australian National Health and Medical Research Council (NHMRC) Australia Fellowship 1062470 to T.R.M., APP1108254 to B.S.K., and APP1114016 to J.B. I.W.D. is supported by a Cancer Institute NSW Early Career Fellowship (2018/ECF013). T.R.M. and T.W. are supported by the Paramor Family Fellowship. S.A.H. is supported by an Australian Postgraduate Award scholarship. The authors would like to thank Dr. A. Verges (UNSW Sydney, Australia) for her supervision and support for N.S.S. The contents of the published material are solely the responsibility of the administering institution, a participating institution, or individual authors, and do not reflect the views of NHMRC.

## Author contributions

S.A.H. and T.R.M. designed sequins, mixtures, and experiments. W.Y.C. and B.S.K. prepared sequins mixtures and libraries for Illumina sequencing. M.A.S. performed MinION library preparation, sequencing, and base calling. S.A.H., T.W., and I.W.D. carried out bioinformatics analysis. S.E.O. and E.M. contributed materials. B.A.N., C. E.L., and L.K.N contributed samples and provided supervision. N.S.S. collected salt-marsh samples and performed DNA extractions. S.A.H. and T.R.M. wrote the manuscript.

## Additional information

**Competing interests:** Author T.R.M. declares the following competing interests: The Garvan Institute of Medical Research has filed a patent (PCT/AU2015/050797) covering aspects of sequencing controls. The remaining authors declare no competing interests.

