## [Peer Review File · Nature Communications]

Reviewers' comments:

Reviewer #1 (Remarks to the Author):

In this manuscript, Hardwick et al developed and validated the utility of 86 synthetic internal DNA standards ('sequins' = sequencing spike-ins) totaling ~230 kb of synthetic microbial community DNA, that are added to environmental DNA for library prep and shotgun metagenome sequencing. The sequins were benchmarked using DNA from a defined mock community of 26 microbial organisms with complete reference genomes and then applied to the wild (i.e. metagenome DNA from saltmarsh samples). The authors illustrate the utility of sequins in measuring fold-change differences in community size / structure and quantitatively normalizing between different samples. Using the spike-ins to normalize between different metagenomics samples, to be used for comparative metagenomics, is specifically exciting. Lastly and forward looking, the authors show how their spike-ins can be used for benchmark testing of new (third generation) sequencing methodologies such as Oxford nanopore technology. Authors provide the physical sequins along with the software tools for free for non-profit research.

While this is a very methods-centric paper, the authors apply sequins to metagenomic DNA and show the application (and potential benefit and utility) in real environmental samples. Sequins are a long-needed addition in the shotgun metagenomic toolkit and will be of great value to the research community, specifically as amplicon sequencing will slowly be replaced through shotgun metagenomics and comparative shotgun metagenomics will become more prevalent and increase in scale, as facilitated through higher throughput sequencing machines such as the Illumina NovaSeq. The methodology and the validation are thorough and robust. Having access to validated and the free of charge metagenome reference standards provides a very valuable resource for microbial ecology and shotgun metagenome studies to come.

MAJOR COMMENTS

A major concern relates to the general writing of the manuscript, specifically the introduction and discussion. For a journal like Nature Communications, one might expect a broader introduction (bird's eye view) leading into the topic of shotgun metagenomics and specifically comparative metagenomics, as this is where much of the sequin utility will come to play. I would suggest to rewrite a small part at the beginning of the introduction to talk about comparative metagenomics and its past and likely future impact on and benefits to the fields of microbial ecology and environmental genomics more broadly. Then one can move into the challenges and what has hampered the field, incl. lack of reference standards, normalization etc..

Likewise, I find the discussion rather weak, esp. for the Nature Communications readership, missing the broader big picture and impact of this approach to microbial ecology and genomics. A discussion with future outlook might be useful too, to discuss if this is something that might be adoptable to for example metatranscriptomics and what the challenges (but also opportunities) might be. As sequins were benchmarked with ONT, a discussion on long read technologies, such as ONT for shotgun metagenomics is warranted.

The comparative genomic analysis of soil, rhizosphere and root warrants more details and discussion. The authors note that without normalization, samples cluster (PCA analysis) by sample type (not environment), though the relative log expression (RLE) box plots show need for normalization. After sequin-based normalization (samples now centered around zero) the authors note that the samples still clustered by sample type. This is great to know and lends some level of support that comparative analysis of previous metagenome studies might be robust even without normalization. Looking at the PCA plots there do seem to be some differences between not normalized and normalized (ie soil and rhizosphere clustering versus some rhizosphere and root clustering). These data should be better explained and discussed, and some statistics should be

applied, incl for sample type reproducibility/ replicates.

MINOR COMMENTS

- The authors provide access to the validated metagenome reference standards free of charge to the research community. Will this service continue and is it sustainable for extended periods of time?

- "have proven useful reference standards for benchmarking different technologies, assessing biases, and for optimizing new analytical methods for metagenomics"

Rinke et al (<https://peerj.com/articles/2486/>) and Bowers et al (<https://www.ncbi.nlm.nih.gov/pubmed/26496746>) references would be suitable references to include here.

- "diverse selection of finished microbial genomes (RefSeq14) that encompassed a wide representation of taxa (including Eukaryota, Bacteria and Archaea), size (0.5-10 Mb), GC content (20-70%), rRNA operon count (1-10) and isolation from a diverse range of environments (human body, aquatic, terrestrial and extreme physical or chemical conditions) (Supplementary Fig. 1)."

How exactly were these selected? Hand-picked to represent a spread? S1 shows a nice spread of phylogenetic diversity and GC. It would be helpful if the figure would also show genome size for each selected taxon, # of rRNA operons and isolation source. These could be easily added as a heat map/ color band and would be very informative and helpful for the reader.

- "A representative section of each genome was selected and inverted to remove homology to any known natural sequences, whilst maintaining nucleotide composition, repeat content and higher order genetic features"

What are the chances that any of the pieces of DNA do have true homology to certain metagenome datasets (not yet in the public database). Even with no hits to the current NCBI nt database, these sequence snippets may be out there. How exactly were they selected in the respective genomes? Authors should provide more details on the exact selection of the sequences. What does "representative section" mean?

- "We next aligned the neat sequins library to a combined index comprising sequins and MBarcode-26 genomes, finding that only a negligible fraction of the reads (0.26%) aligned to an MBarcode-26 genome (Supplementary Fig. 2b). Close inspection of these cross-aligning reads indicated they likely derive from microbial contamination in laboratory reagents."

Please specify/ provide more details.

- "Of the resulting library, 1.49% of reads aligned to sequins, while 98.1% aligned to MBarcode-26 genomes."

What about the remaining ~0.4%? Please explain.

- "Long-read sequencing can span and anchor repetitive elements within a microbial genome, and is useful for genome assembly"

Suggest rewording; it's useful for high quality genome assembly as it resolves repeats..

- Indel rate in polyN stretches in ONT data: "MinION sequencing notably suffered high indel rates in homopolymeric sequences with characteristic sequence coverage drops at the upstream end of homopolymer tracts (Supplementary Fig. 5c)."

Instead (or in addition) of the genome browser view in 5C that shows an example of a homopolymer stretch, it might be more useful to have a general quantification of the #/% of indels that are in homopolymeric sequences.

- "whilst the N50 values for sample contigs were poorer (697 - 3,659 kb), indicating poor sample quality limited assembly, rather than technical factors."

Poor sample quality may not be the correct term here - please reword.

Reviewer #2 (Remarks to the Author):

Overview:

Hardwick et al. develop and validate synthetic spike-in standards (which the authors call sequins) for metagenomic sequencing studies, relying heavily on similar approaches developed for RNA expression analysis. Importantly, the collection of 86 sequences used as spike-ins are either random sequences or reversed sequences found in existing genomes, and so have no known homology to known DNA sequence. This design allows for use of the spike-ins for accurate quantification and normalization of abundance estimates, without the risk that assembly and quantification of the spike ins will be confounded by the presence of similar sequence in any given metagenomic sample. The authors carefully and methodically validate the utility of this approach in several controlled experiments and example case-studies. Most notably, the authors use the spike-ins to benchmark error rates of sequencing technologies, and assembly and mapping protocols by coverage; to normalize abundance estimates of individual genomes from sample to sample; and to quantify changes in absolute abundance for genomes across samples using fixed spike-in masses. In general, the experiments are carefully thought out and well-controlled, without being overly simplistic.

The authors carefully address a pressing need in the field to make more robust quantitative estimates of genome abundance in metagenomics studies. This is especially timely given the increasing adoption of shotgun metagenomics as a tool for making conclusions based on comparative abundance of genomes across samples. Sequins are not a panacea for all the complexities of such approaches. For example, they do nothing to address the complexities of making abundance estimates in the face of (variable) strain variation. However, I think this work is an important step in the right direction, and this is a nice resource for the community. I am unaware of similar resources.

The manuscript is clearly written and logically organized, with minor exceptions noted below.

Major Comments:

1. Figure 4 - The ability to measure absolute changes in abundance across samples is one of the most intriguing results of the paper. However, there are some confusing parts to this figure.
 - 1.1. First, I believe the abundances of at least *Synechocystis* sp. PCC 6803 may have errors in Panel 4a (or elsewhere). For example, the abundance in Mixture A3 is equal to the abundance in mixture B1 in this panel, yet panel d shows expected fold-changes (0.5) that do not reflect this. As another example, A2 / A3 is also inconsistent between panel a and panel d, if I am interpreting this figure as intended.

1.2. Second, it is confusing to have the "(e.g. 10ng)" label on the figure. That would represent a relatively large bar in panel a, and appears not to be what the authors actually spiked in based on panel b and methods (2/3 ng).

1.3. Panel c does not match Panel a exactly, and based on the great result in panel e I further suspect panel a is incorrect. Also, is there any reason the authors use units of "normalized signal" here instead of directly calculating ng of DNA for each of the Cyanobacteria mock community members? Unless I'm missing something, it seems like this should be possible.

2. Figure 3 (and associated text)- I'm not I understand the point of the comparison of PCA plots. The authors state minimally that "samples still clustered by sample type," but I am left eyeballing the plots and noting that the statement is not strictly true for either of the plots. The individual relationships between pairs of samples appear to have possibly shifted dramatically, though it is impossible to say without labeling individual replicates (look at the root samples for an example of this). That is, the beta diversity relationships appear to have changed, which is not what the authors highlight in calling out the consistent clustering by sample type. Can the authors expand on what I am supposed to take away from this result? If I am to take away from the RLE plots that normalization is successful, and I will have more accurate quantification of phylum-level abundances, then I would expect the PCA plots to be meaningfully different pre and post normalization. I am left wondering what the "correct" PCA plot should look like, based on evidence or at least some expanded justification from the authors. A related concern is the very use of RLE plots. Given that these were from fairly different sample types, and that this data appears to have been summarized at the phylum level (many fewer phyla than genes in a gene expression scenario), is it appropriate to assume that most phyla will not be expected to change from sample to sample?

3. Methods, page 12, design: Sequin design could be fleshed out a bit. What does "if necessary, shuffled" mean? Were genomes shuffled or sequins shuffled? I'm not shure how you shuffle a genome and maintain repeats and distribution of k-mers, for example. How were the sequins selected from the larger genomes from which they were sampled? What does "checked against the BLAST nt database" mean?

4. In Supplemental Figure 4b, I don't understand why the authors are normalizing by mean coverage of all sequins in a staggered community (the same could be said for MBarC-26) in order to show there is no GC bias for coverage. Many dots fall at zero on the y-axis, but is this not simply because many sequins have very low expected coverage compared to the mean? To make the point that GC content does not affect measured coverage, something else should be plotted, for example GC content vs. observed / expected coverage.

5. Results, page 4: "Close inspection of these cross-aligning reads indicated they likely derive from microbial contamination in laboratory reagents" It's not clear to me what the evidence for this claim is, and I couldn't find what "close inspection" meant in the methods.

6. Sequin length varies substantially, and several are ≤ 1000 bp. Read mapping /coverage will be diminished near the ends of sequins. Is there a relationship between sequin length and observed / expected coverage?

7. Results, Page 7: "whilst the N50 values for sample contigs were poorer (697 - 3,659 b), indicating poor sample quality limited assembly, rather than technical factors." First, small typo in units for N50. The authors above indicate that many non-technical factors can affect assembly ("complexity of samples, the presence of repeats and closely related strains, and the presence of many species at low abundance"). However, at the end of this paragraph, they declare that good sequin assembly means that "poor sample quality limited assembly." First, I'm not sure what poor sample quality means, here. Can a more precise definition be used? Second, what about the other factors (complexity, strains, long tail distribution) that they mention above? These seem more important limitations to assembly, and one that I don't think sequins will address. In short, I think it's important to be clear here what spike-ins can tell you and what they can't; the language in this section is important.

Minor Comments:

1. Abstract: "developed a set internal DNA"  developed a set of internal DNA"

2. Abstract: "accompanying microbe communities"  "accompanying microbial communities"

3. Introduction, final paragraph: I think the phrase "that represents a collection of natural microbial genomes" is confusing this early in the paper. I think adding a sentence here explaining that these are sequences *not* found in "natural microbial genomes" would help the reader grasp earlier on what you are doing. I'm not sure what the right phrase here is, but this sentence as written had me thinking the synthetic sequences were sampled directly from genomes the first time I read it.
4. Results, page 2: It's unclear what "higher order genetic features" means.
5. Fig 1d, 1e: how were dashed lines fit?
6. Fig 2c (and elsewhere): I suggest using "sequin assembly (%)" on the y-axis. In the legend, you use the phrase, "fraction of each sequin de novo assembled" and I think this makes it much clearer and suggest you be consistent in axis label with calling these sequins in y axis. What are error bars in this plot?
7. Fig 3 legend: "between replicates"  among replicates
8. Results, page 6: The high indel rates near homopolymers for the MinION sequencing is an interesting observation, as is the fact that indels are not symmetrically distributed across the homopolymer. Has this been observed before? If so, would be nice to point that out. If this is novel, that would also be nice to point out.
9. Results, page 7: "assembly is due to sampe complexity"  ... sample complexity
10. Some sequins are outliers in de novo assembly coverage (Fig. 1) and error rate (Fig S4). Do the authors recommend all sequins for use, or did they remove any designed sequins from what they are making available?

Synthetic microbe communities provide internal reference standards for metagenome sequencing and analysis.

Hardwick, S.A. *et al.*

We thank both reviewers for contributing their time to review the manuscript, and we believe the manuscript has improved as a result of their useful feedback and thoughts. We have addressed each specific reviewer comment below. All changes to the manuscript are marked up in this document and the manuscript itself.

Reviewer #1:

Remarks to the Author:

In this manuscript, Hardwick et al developed and validated the utility of 86 synthetic internal DNA standards ('sequins' = sequencing spike-ins) totaling ~230 kb of synthetic microbial community DNA, that are added to environmental DNA for library prep and shotgun metagenome sequencing. The sequins were benchmarked using DNA from a defined mock community of 26 microbial organisms with complete reference genomes and then applied to the wild (i.e. metagenome DNA from saltmarsh samples). The authors illustrate the utility of sequins in measuring fold-change differences in community size / structure and quantitatively normalizing between different samples. Using the spike-ins to normalize between different metagenomics samples, to be used for comparative metagenomics, is specifically exciting. Lastly and forward looking, the authors show how their spike-ins can be used for benchmark testing of new (third generation) sequencing methodologies such as Oxford nanopore technology. Authors provide the physical sequins along with the software tools for free for non-profit research.

While this is a very methods-centric paper, the authors apply sequins to metagenomic DNA and show the application (and potential benefit and utility) in real environmental samples. Sequins are a long-needed addition in the shotgun metagenomic toolkit and will be of great value to the research community, specifically as amplicon sequencing will slowly be replaced through shotgun metagenomics and comparative shotgun metagenomics will become more prevalent and increase in scale, as facilitated through higher throughput sequencing machines such as the Illumina NovaSeq. The methodology and the validation are thorough and robust. Having access to validated and the free of charge metagenome reference standards provides a very valuable resource for microbial ecology and shotgun metagenome studies to come.

We thank Reviewer 1 for their encouraging summary of our work.

MAJOR COMMENTS

A major concern relates to the general writing of the manuscript, specifically the introduction and discussion. For a journal like Nature Communications, one might expect a broader introduction (bird's eye view) leading into the topic of shotgun metagenomics and specifically comparative metagenomics, as this is where much of the sequin utility will come to play. I would suggest to rewrite a small part at the beginning of the introduction to talk about comparative metagenomics and its past and likely future

impact on and benefits to the fields of microbial ecology and environmental genomics more broadly. Then one can move into the challenges and what has hampered the field, incl. lack of reference standards, normalization etc..

We have now expanded the scope of the introduction to discuss the benefits and impact of shotgun and comparative metagenomics on microbial ecology and environmental genomics as follows:

“The sequencing of DNA recovered directly from environmental samples can reveal the presence of microbial communities without requiring prior laboratory cultivation^{1,2}. This approach, termed metagenomics, has exposed previously hidden microbial diversity in a range of different environments, from the open ocean³, to complex soil samples⁴, to the human microbiome⁵. Accordingly, metagenomics is often used to determine the profile of the microbes that inhabit a given environment, to diagnose the presence of a microbial pathogen⁶ or identify novel microbial lineages⁷.

Comparisons between microbial communities that inhabit different environmental sites can also distinguish differences in the identity and abundance of microbes⁸. These approaches can identify microbes that confer specific environmental characteristics, or measure the impact of environmental variables on microbial communities, and have been used to discover host-microbe interactions, identify novel microbes with biotechnological value, and measure environmental health⁹.

Despite the promise of this approach, the analysis of metagenomics data remains challenging. The sheer size and complexity of microbial genomes within a sample, many of which may be novel, confounds reliable identification and makes quantification of microbes difficult¹. Additional technical biases that accrue during next-generation sequencing (NGS) further bias metagenomic analysis². Differences in microbial population structures can also invalidate the assumptions that underlie normalization approaches, and thereby preclude accurate detection of genuine biological differences between samples⁸.

Reference standards can offset these analytical challenges^{10,11}. Reference standards enable the limits of sampling and analysis to be understood and can measure technical variables that bias analysis with NGS. Reference standards can also evaluate quantitative accuracy, and act as scaling factors by which to normalize between samples. Accordingly, there is a pressing need to develop metagenome reference standards with known properties that can be used to benchmark analytical methods and enable comparisons between multiple samples.”

Likewise, I find the discussion rather weak, esp. for the Nature Communications readership, missing the broader big picture and impact of this approach to microbial ecology and genomics. A discussion with future outlook might be useful too, to discuss if this is something that might be adoptable to for example metatranscriptomics and what the challenges (but also opportunities) might be. As sequins were benchmarked with ONT, a discussion on long read technologies, such as ONT for shotgun metagenomics is warranted.

Likewise, we have now expanded the discussion to incorporate these additional aspects as follows:

“Metagenomics can profile the community of microbes within an environmental sample. This analysis of microbes does not require prior laboratory cultivation, can discover new microbial lineages⁷, and diagnose the presence of pathogens within patient samples⁶.

Despite this promise, the analysis and comparison of metagenome samples is challenging, and reference standards are needed to ensure the accuracy and reproducibility of results¹¹.

Here, we describe the development of reference standards, termed 'sequins', for metagenome experiments. Sequins comprise a set of synthetic DNA standards that mimic the sequence complexity, phylogenetic composition and GC content of a natural microbial community, yet have no homology to known nucleic acid sequences. This enables their use as internal reference standards for downstream steps, including library preparation, sequencing and bioinformatic analysis.

Metagenome samples can be affected by a range of unwanted technical variables, such as different library preparation methods⁴² sequencing technologies, or the presence of enzymatic inhibitors⁴³. A major advantage of sequins is their ability to measure and mitigate this technical variation that influences sequencing. As internal standards, sequins can be used to normalize different methods, experiments and batches, and distinguish genuine biological differences from confounding technical variables³⁹.

As well as enabling quality control and inter-sample normalization, reference standards are also essential for the development and optimization of new sequencing technologies^{11,14}. In this study, we used sequins to benchmark the performance of both short-read and long-read sequencing technology, as well as a range of different software tools. Emerging long-read technologies offer several advantages for metagenomics, including more accurate resolution of repeat elements³², and the ability to carry out real-time, portable genomic surveillance in the field during disease outbreaks⁴⁴. As new technologies continue to develop, sequins provide a constant reference by which to benchmark different tools.

Global differences in the size and complexity of microbial populations can violate the assumptions that underlie normalization between samples. For example, the comparison of the taxonomic sub-populations within a microbial community is limited to the relative abundance of taxa, and metagenomics cannot detect shifts in absolute abundance between samples⁸. Here, we demonstrate how sequins can be used to calibrate and rescale different samples, thereby enabling the accurate measurement of both relative and absolute fold-changes.

Metagenome studies often sample widely disparate environmental sites, and often at different times and methods, resulting in substantial batch variation between experimental samples. We propose that the routine use of reference standards, such as sequins, provides a reliable reference against which to standardize sample collection across such different sites, and can support more rigorous meta-analysis between different experiments and studies.

Within the laboratory, sequins can be used for the routine surveillance of individual samples, and to measure and improve operational performance. This is particularly important in clinical microbiology, where equipment and processes must be regularly validated, and the diagnosis of pathogens within patient samples assured^{6,12}. Indeed, we expect that reference standards such as sequins will be useful for clinical assay development and validation, and also for ongoing quality control and proficiency testing procedures^{6,45}.

Given these benefits to metagenome methods, we have made sequins, along with associated datasets, protocols and an accompanying software toolkit, freely available to researchers on our website at www.sequin.xyz.

The comparative genomic analysis of soil, rhizosphere and root warrants more details and discussion. The authors note that without normalization, samples cluster (PCA analysis) by sample type (not environment), though the relative log expression (RLE) box plots show need for normalization. After sequin-based normalization (samples now centered around zero) the authors note that the samples still clustered by sample type. This is great to know and lends some level of support that comparative analysis of previous metagenome studies might be robust even without normalization. Looking at the PCA plots there do seem to be some differences between not normalized and normalized (ie soil and rhizosphere clustering versus some rhizosphere and root clustering). These data should be better explained and discussed, and some statistics should be applied, incl for sample type reproducibility/replicates.

We have added a new panel (**Fig. 3b**) that compares the coefficient of variation (standard deviation divided by mean) between replicates before and after RUVg normalization. This shows that the mean coefficient of variation decreases for all six sample groups after RUVg normalization, illustrating the benefit of RUVg normalization using the sequins.

This additional analysis, and expanded explanation has been included in the results as follows:

*“We next examined the taxonomic composition of reads using MG-RAST²¹, by searching against the RefSeq database (phylum level). Principal component analysis (PCA) revealed that samples clustered loosely by type (non-rooted soil, rhizosphere, root) rather than environment (natural, regenerated) (**Fig. 3c, left**). However, box plots of relative log expression (RLE)²² showed a clear need for normalization, with large distributional differences between samples that indicated unwanted variation (**Fig. 3c, right**). RLE plots, while originally devised for gene expression data, can also be used to reveal unwanted variation in many other kinds of high-dimensional data, and are particularly useful for assessing whether a normalization procedure has been successful²².”*

*To normalize between samples, we employed the RUVg method, which performs factor analysis on suitable sets of control genes (e.g. spike-in controls) to adjust for unwanted technical variation¹⁴. To perform RUVg normalization, we nominated the subset of sequins that remain at equimolar concentrations across mixtures as negative controls (n=36). RUVg normalization improved the data, with samples still clustering loosely by type (**Fig. 3d, left**), and sample RLE plots now centered around zero with most of the excessive variation removed (**Fig. 3d, right**). While samples still did not cluster perfectly by type, it must be borne in mind that the replicates were collected from three separate natural and regenerated sites. Nonetheless, the fact that samples clustered loosely by type but seemingly not at all by environment is an interesting biological observation, and implies that regeneration of these saltmarsh sites after initial development did not have a discernible effect on the microbial ecology of the samples. In order to compare variation between replicates, we plotted the mean coefficient of variation for each of the six treatment groups both before and after RUVg normalization, finding that variation decreased substantially after RUVg normalization (**Fig. 3b**). This provided further evidence that RUVg normalization was successful. Notably, the use of sequins with the RUVg approach outperformed other normalization methods, including upper-quartile (UQ) normalization, where some samples still displayed excessive variability (see **Supplementary Fig. 8 and 9**).”*

MINOR COMMENTS

- The authors provide access to the validated metagenome reference standards free of charge to the research community. Will this service continue and is it sustainable for extended periods of time?

We plan to provide sequins to the non-profit research community free of charge indefinitely. Given the research was publicly funded and the Garvan Institute is a non-profit institution, we consider it appropriate that the sequins are provided as a free resource for the non-profit research community. Indeed, it is our aim that sequins are used as broadly as possible for research use.

- "have proven useful reference standards for benchmarking different technologies, assessing biases, and for optimizing new analytical methods for metagenomics" Rinke et. al., (<https://peerj.com/articles/2486/>) and Bowers et al (<https://www.ncbi.nlm.nih.gov/pubmed/26496746>) references would be suitable references to include here.

We have now cited these references to support this statement in the manuscript.

- "diverse selection of finished microbial genomes (RefSeq14) that encompassed a wide representation of taxa (including Eukaryota, Bacteria and Archaea), size (0.5-10 Mb), GC content (20-70%), rRNA operon count (1-10) and isolation from a diverse range of environments (human body, aquatic, terrestrial and extreme physical or chemical conditions) (Supplementary Fig. 1)."

How exactly were these selected? Hand-picked to represent a spread? S1 shows a nice spread of phylogenetic diversity and GC. It would be helpful if the figure would also show genome size for each selected taxon, # of rRNA operons and isolation source. These could be easily added as a heat map/color band and would be very informative and helpful for the reader.

The microbial genomes were ranked according to the range of genome features, and then manually curated in order to ensure a proportional representation of each of these features. We have now updated **Supplementary Table 1** to include RefSeq accession ID, genome size, number of 16S rRNA copies and isolation source for each sequin. We have also amended **Supplementary Fig. 1** to indicate genome size and 16S rRNA copy number.

- "A representative section of each genome was selected and inverted to remove homology to any known natural sequences, whilst maintaining nucleotide composition, repeat content and higher order genetic features"

What are the chances that any of the pieces of DNA do have true homology to certain metagenome datasets (not yet in the public database). Even with no hits to the current NCBI nt database, these sequence snippets may be out there. How exactly were they selected in the respective genomes? Authors should provide more details on the exact selection of the sequences. What does "representative section" mean?

Whilst we cannot rule out the possibility that sequins may share some sequence homology with novel microbes that are not yet described in public databases, the chance that this homology would extend across the entire read length, and therefore result in erroneous cross-alignment, is exceedingly low. Nevertheless, we have taken prudent steps to minimize the chances of this happening, including (i) pre-

screening all potential sequences against the entire BLAST nt database, and (ii) preparing and sequencing neat mixtures of sequins (i.e. with no natural DNA added) and testing if sequencing reads derived from sequins (which contain sequencing errors) return any positive hits to the nt database using Centrifuge.

We have also provided the following additional details on the selection of sequin sequences as follows:

"To design artificial sequences, we first retrieved all high quality, finished, microbial genome sequences from GenBank²³. We then ranked and systematically selected genomes according to a range of features, including taxa (including representatives from Eukaryota, Bacteria and Archaea), size (0.5–10 Mb for prokaryotes), GC content (20–71%), rRNA operon count (1–11), and isolation from a diverse range of environments (human body, aquatic, terrestrial and extreme physical or chemical conditions) (Supplementary Fig. 1). This ensured that the selected genomes provided a proportional representation of these different features. To illustrate the evolutionary relationships between these microbial genomes, we generated a phylogenetic tree using iTOL (v3) with default parameters²⁴.

We then aimed to select a sub-sequence representative of each genome as follows. We first selected a large number (>100) of random ~1-10 kb sub-sequences of each genome. Sub-sequences were then inverted to remove primary sequence homology, whilst still retaining different features of the original microbial genome, e.g. size, nucleotide composition, GC content, and the distribution of repetitive and unique sequences. Inverted sequences were then queried against the BLAST nt database for homology to known sequences and any inverted sequence with a significant match (E-value < 0.01) was omitted from further analysis. From the remaining sub-sequences, we then selected a sub-sequence that had a GC content closest to the original source genome. In addition, some sequences were manually curated to generate sequins with extremely high or low GC content. Upon completion, we generated a set of 86 synthetic DNA standards ('sequins') which ranged in length from ~1–10 kb, with a combined size of ~227 kb (see Supplementary Table 1 for full details of sequences)."

- "We next aligned the neat sequins library to a combined index comprising sequins and MBARC-26 genomes, finding that only a negligible fraction of the reads (0.26%) aligned to an MBARC-26 genome (Supplementary Fig. 2b). Close inspection of these cross-aligning reads indicated they likely derive from microbial contamination in laboratory reagents."

Please specify/ provide more details.

The vast majority (~99%) of these cross-aligning reads aligned to the *E. coli* genome (NC_000913.3), which has previously been found to be a common source of microbial contamination in laboratory reagents^{25,26}. Therefore, we conclude it is likely that these reads represent microbial contamination that has been introduced in laboratory reagents or processes. Therefore, we have amended the text as follows:

*"We next aligned the neat sequins library to a combined index comprising sequins and MBARC-26 genomes, finding that only a negligible fraction of the reads (0.26%) aligned to an MBARC-26 genome (Supplementary Fig. 2b). The vast majority (>99%) of these cross-aligning reads aligned to the *E. coli* K-12 genome (NC_000913.3), and likely result from contamination in laboratory*

reagents and processes^{25,26}."

- "Of the resulting library, 1.49% of reads aligned to sequins, while 98.1% aligned to MBarC-26 genomes."

What about the remaining ~0.4%? Please explain.

The remaining ~0.4% of reads did not align to any sequin or MBarC-26 genome. Again, it is likely these non-aligned reads result from sequencing errors or failures and unidentified contamination.

- "Long-read sequencing can span and anchor repetitive elements within a microbial genome, and is useful for genome assembly"

Suggest rewording; it's useful for high quality genome assembly as it resolves repeats..

We have now reworded this sentence as follows:

"Long-read sequencing can resolve repetitive regions within a microbial genome, and is useful for genome assembly¹⁷."

- Indel rate in polyN stretches in ONT data: "MinION sequencing notably suffered high indel rates in homopolymeric sequences with characteristic sequence coverage drops at the upstream end of homopolymer tracts (Supplementary Fig. 5c)."

Instead (or in addition) of the genome browser view in 5C that shows an example of a homopolymer stretch, it might be more useful to have a general quantification of the #/% of indels that are in homopolymeric sequences.

We agree that it would be useful to provide some quantification of the number of indels in homopolymers. Accordingly, we have done a global analysis of indel rates in the MinION data and revised the text as follows:

"We next measured sequencing error rates, finding that MinION had a mismatch error rate of 7.12% (compared to 0.127% for Illumina) and an indel rate of 8.71% (compared to 0.00770% for Illumina). Notably, MinION sequencing suffered significantly higher indel rates in homopolymeric sequences (mean indel rate = 16.7%) compared to other regions (mean = 7.69%; unpaired t-test with Welch's correction, p-value < 0.0001), with characteristic sequence coverage drops at the upstream end of homopolymer tracts (Supplementary Fig. 5c). This phenomenon has also been reported by others^{17,27,28}."

- "whilst the N50 values for sample contigs were poorer (697 - 3,659 kb), indicating poor sample quality limited assembly, rather than technical factors."

Poor sample quality may not be the correct term here - please reword.

We agree and have reworded this sentence as follows:

*“De novo assembly of metagenome samples can be confounded by both inherent sample complexity, as well as technical factors introduced during library preparation, sequencing or analysis. Sequins can distinguish between these outcomes and thereby assist in quality control and troubleshooting. To demonstrate this, we performed de novo assembly on all samples using Ray Meta, and evaluated the quality of each assembly using MetaQUAST (supplying the sequin reference sequences). The N50 values for contigs aligned to sequins (2,272 – 2,879 nt) indicated that library preparation, sequencing and bioinformatic assembly were performed successfully (**Supplementary Fig. 7c**). Conversely, the N50 values for unaligned contigs were poorer and less consistent (697 – 3,659 nt), indicating that inherent sample complexity limited assembly, rather than technical factors.”*

Reviewer #2 (Remarks to the Author):

Overview:

Hardwick et al. develop and validate synthetic spike-in standards (which the authors call sequins) for metagenomic sequencing studies, relying heavily on similar approaches developed for RNA expression analysis. Importantly, the collection of 86 sequences used as spike-ins are either random sequences or reversed sequences found in existing genomes, and so have no known homology to known DNA sequence. This design allows for use of the spike-ins for accurate quantification and normalization of abundance estimates, without the risk that assembly and quantification of the spike ins will be confounded by the presence of similar sequence in any given metagenomic sample. The authors carefully and methodically validate the utility of this approach in several controlled experiments and example case-studies. Most notably, the authors use the spike-ins to benchmark error rates of sequencing technologies, and assembly and mapping protocols by coverage; to normalize abundance estimates of individual genomes from sample to sample; and to quantify changes in absolute abundance for genomes across samples using fixed spike-in masses. In general, the experiments are carefully thought out and well-controlled, without being overly simplistic.

The authors carefully address a pressing need in the field to make more robust quantitative estimates of genome abundance in metagenomics studies. This is especially timely given the increasing adoption of shotgun metagenomics as a tool for making conclusions based on comparative abundance of genomes across samples. Sequins are not a panacea for all the complexities of such approaches. For example, they do nothing to address the complexities of making abundance estimates in the face of (variable) strain variation. However, I think this work is an important step in the right direction, and this is a nice resource for the community. I am unaware of similar resources.

The manuscript is clearly written and logically organized, with minor exceptions noted below.

We thank Reviewer 2 for their kind appraisal of our study.

Major Comments:

1. Figure 4 - The ability to measure absolute changes in abundance across samples is one of the most intriguing results of the paper. However, there are some confusing parts to this figure.

1.1. First, I believe the abundances of at least *Synechocystis* sp. PCC 6803 may have errors in Panel 4a (or elsewhere). For example, the abundance in Mixture A3 is equal to the abundance in mixture B1 in this panel, yet panel d shows expected fold-changes (0.5) that do not reflect this. As another example, A2 / A3 is also inconsistent between panel a and panel d, if I am interpreting this figure as intended.

We thank the astute reviewer for pointing out this error regarding comparisons between the A and B mixtures which has now been corrected in the figure. However, the abundances for A2/A3 were correct; *Synechocystis* sp. PCC 6803 doubles in abundance from A2 to A3, and this is reflected in **Fig. 4d**.

1.2. Second, it is confusing to have the "(e.g. 10ng)" label on the figure. That would represent a relatively large bar in panel a, and appears not to be what the authors actually spiked in based on panel b and methods (2/3 ng).

We apologies for the confusion resulting from the inclusion of “eg. 10ng”, and this has been removed from the figure.

1.3. Panel c does not match Panel a exactly, and based on the great result in panel e I further suspect panel a is incorrect. Also, is there any reason the authors use units of "normalized signal" here instead of directly calculating ng of DNA for each of the *Cyanobacteria* mock community members? Unless I'm missing something, it seems like this should be possible.

The reviewer is correct that the re-scaled abundances in **panel (c)** do not exactly match **panel (a)**. Nevertheless, whilst not perfect, these two panels aims to illustrate that normalization using sequins is more faithful to the true microbial composition than performing standard normalization (solely based on genome size and library depth). As requested, we have now converted “normalized signal” in **panel (c)** to mass of DNA (ng), by reference to the known input amount of sequins.

2. Figure 3 (and associated text)- I'm not I understand the point of the comparison of PCA plots. The authors state minimally that "samples still clustered by sample type," but I am left eyeballing the plots and noting that the statement is not strictly true for either of the plots. The individual relationships between pairs of samples appear to have possibly shifted dramatically, though it is impossible to say without labeling individual replicates (look at the root samples for an example of this). That is, the beta diversity relationships appear to have changed, which is not what the authors highlight in calling out the consistent clustering by sample type. Can the authors expand on what I am supposed to take away from this result? If I am to take away from the RLE plots that normalization is successful, and I will have more accurate quantification of phylum-level abundances, then I would expect the PCA plots to be meaningfully different pre and post normalization. I am left wondering what the "correct" PCA plot should look like, based on evidence or at least some expanded justification from the authors. A related concern is the very use of RLE plots. Given that these were from fairly different sample types, and that this data appears to have been summarized at the phylum level (many fewer phyla than genes in a gene expression scenario), is it appropriate to assume that most phyla will not be expected to change from sample to sample?

We thank the reviewer for this constructive feedback. We have now labeled the individual replicates in **Fig. 3** to enable tracking of individual samples before/after normalization. While it is true that the saltmarsh samples don't cluster perfectly by type, we were simply making the point that they cluster loosely by type (they certainly cluster more strongly by type than environment). This loose clustering is maintained after performing RUVg normalization using sequins, which suggests that the biological signal has been retained while much unwanted variation has been removed (as evidenced by the RLE plots).

This is an interesting observation, as it suggests that regeneration of these salt marsh environments after initial development did not have a discernible effect on the microbial ecology of the samples. Unfortunately we don't know what the “correct” PCA plot looks like, as we are profiling real metagenome samples of unknown content.

We have also taken on board the reviewer's broader concern about the use of RLE plots. RLE plots, while originally devised for gene expression data, can also be used to reveal unwanted variation in many other kinds of high-dimensional data, e.g. metabolomic and proteomic data²². While we are not aware of RLE plots being applied to metagenomic data previously, it is in principle no different from other kinds of high-dimensional sequencing data, and the RLE plots provide a useful illustration to assess whether a

normalization procedure has been successful. In support of this position, previous metagenomic studies have made the assumption that the majority of taxonomic groups are not differentially abundant between samples (see e.g. Sohn *et al.* 2015²⁹). We concede that the assumption may not hold true for some extreme cases, e.g. where antibiotic treatment can produce profound changes of microbial compositions in human samples (e.g. David *et al.* 2014³⁰).

We have also now added a new panel **(b)** to **Fig. 3**, which compares the coefficient of variation (standard deviation divided by mean) between replicates before and after RUVg normalization. It shows that mean coefficient of variation decreases substantially for all samples after RUVg normalization, providing further evidence that RUVg normalization has been successful.

We have amended the Results section as follows:

*“We next examined the taxonomic composition of reads using MG-RAST²¹, by searching against the RefSeq database (phylum level). Principal component analysis (PCA) revealed that samples clustered loosely by type (non-rooted soil, rhizosphere, root) rather than environment (natural, regenerated) (**Fig. 3c, left**). However, box plots of relative log expression (RLE)²² showed a clear need for normalization, with large distributional differences between samples that indicated unwanted variation (**Fig. 3c, right**). RLE plots, while originally devised for gene expression data, can also be used to reveal unwanted variation in many other kinds of high-dimensional data, and are particularly useful for assessing whether a normalization procedure has been successful²².”*

*To normalize between samples, we employed the RUVg method, which performs factor analysis on suitable sets of control genes (e.g. spike-in controls) to adjust for unwanted technical variation¹⁴. To perform RUVg normalization, we nominated the subset of sequins that remain at equimolar concentrations across mixtures as negative controls (n=36). RUVg normalization improved the data, with samples still clustering loosely by type (**Fig. 3d, left**), and sample RLE plots now centered around zero with most of the excessive variation removed (**Fig. 3d, right**). While samples still did not cluster perfectly by type, it must be borne in mind that the replicates were collected from three separate natural and regenerated sites. Nonetheless, the fact that samples clustered loosely by type but seemingly not at all by environment is an interesting biological observation, and implies that regeneration of these saltmarsh sites after initial development did not have a discernible effect on the microbial ecology of the samples. In order to compare variation between replicates, we plotted the mean coefficient of variation for each of the six treatment groups both before and after RUVg normalization, finding that variation decreased substantially after RUVg normalization (**Fig. 3b**). This provided further evidence that RUVg normalization was successful. Notably, the use of sequins with the RUVg approach outperformed other normalization methods, including upper-quartile (UQ) normalization, where some samples still displayed excessive variability (see **Supplementary Fig. 8 and 9**).”*

3. Methods, page 12, design: Sequin design could be fleshed out a bit. What does "if necessary, shuffled" mean? Were genomes shuffled or sequins shuffled? I'm not sure how you shuffle a genome and maintain repeats and distribution of k-mers, for example. How were the sequins selected from the larger genomes from which they were sampled? What does "checked against the BLAST nt database" mean?

We have now expanded our description of how sequins are designed from the original microbe genomes:

“To design artificial sequences, we first retrieved all high quality, finished, microbial genome sequences from GenBank²³. We then ranked and systematically selected genomes according to a range of features, including taxa (including representatives from Eukaryota, Bacteria and Archaea), size (0.5–10 Mb for prokaryotes), GC content (20–71%), rRNA operon count (1–11), and isolation from a diverse range of environments (human body, aquatic, terrestrial and extreme physical or chemical conditions) (Supplementary Fig. 1). This ensured that the selected genomes provided a proportional representation of these different features. To illustrate the evolutionary relationships between these microbial genomes, we generated a phylogenetic tree using iTOL (v3) with default parameters²⁴.

We then aimed to select a sub-sequence representative of each genome as follows. We first selected a large number (>100) of random ~1-10 kb sub-sequences of each genome. Sub-sequences were then inverted to remove primary sequence homology, whilst still retaining different features of the original microbial genome, e.g. size, nucleotide composition, GC content, and the distribution of repetitive and unique sequences. Inverted sequences were then queried against the BLAST nt database for homology to known sequences and any inverted sequence with a significant match (E-value < 0.01) was omitted from further analysis. From the remaining sub-sequences, we then selected a sub-sequence that had a GC content closest to the original source genome. In addition, some sequences were manually curated to generate sequins with extremely high or low GC content. Upon completion, we generated a set of 86 synthetic DNA standards (‘sequins’) which ranged in length from ~1–10 kb, with a combined size of ~227 kb (see Supplementary Table 1 for full details of sequences).”

4. In Supplemental Figure 4b, I don't understand why the authors are normalizing by mean coverage of all sequins in a staggered community (the same could be said for MBarC-26) in order to show there is no GC bias for coverage. Many dots fall at zero on the y-axis, but is this not simply because many sequins have very low expected coverage compared to the mean? To make the point that GC content does not affect measured coverage, something else should be plotted, for example GC content vs. observed / expected coverage.

Normalization of sequins was initially performed by mean coverage to account for differences in the abundance of sequins (given the mixtures spiked into samples vary). We have now plotted observed/expected coverage relative to GC in **Supplementary Fig. 4(b)** as requested by the reviewer. We have additionally included the following section in Methods to describe the calculation of observed/expected coverage:

“To assess GC content bias, we calculated observed/expected coverage as the mean coverage of each sequin divided by its expected concentration in the mixture. This controls for the fact that sequin abundances vary over several orders of magnitude within the staggered mixtures”.

5. Results, page 4: "Close inspection of these cross-aligning reads indicated they likely derive from microbial contamination in laboratory reagents" It's not clear to me what the evidence for this claim is, and I couldn't find what "close inspection" meant in the methods.

The vast majority (>99%) of these cross-aligning reads aligned to the *E. coli* genome (NC_000913.3), which has previously been found to be a common source of microbial contamination in laboratory reagents^{25,26}. Thus, we have amended the text as follows:

"We next aligned the neat sequins library to a combined index comprising sequins and MBarcode-26 genomes, finding that only a negligible fraction of the reads (0.26%) aligned to an MBarcode-26 genome (Supplementary Fig. 2b). The vast majority (>99%) of these cross-aligning reads aligned to the E. coli K-12 genome (NC 000913.3), and likely result from contamination in laboratory reagents and processes^{25,26}."

6. Sequin length varies substantially, and several are ≤ 1000 bp. Read mapping /coverage will be diminished near the ends of sequins. Is there a relationship between sequin length and observed / expected coverage?

To illustrate the impact of "edge effects", where read coverage can be diminished at the sequin termini, we plotted observed/expected coverage relative to sequin length (see new panel c of Supplementary Fig. 4). This plot shows evidence for slightly reduced coverage at both the small and large ends of the spectrum, with medium-sized sequins (~2-4 kb) showing slightly higher coverage.

7. Results, Page 7: "whilst the N50 values for sample contigs were poorer (697 - 3,659 b), indicating poor sample quality limited assembly, rather than technical factors." First, small typo in units for N50. The authors above indicate that many non-technical factors can affect assembly ("complexity of samples, the presence of repeats and closely related strains, and the presence of many species at low abundance"). However, at the end of this paragraph, they declare that good sequin assembly means that "poor sample quality limited assembly." First, I'm not sure what poor sample quality means, here. Can a more precise definition be used? Second, what about the other factors (complexity, strains, long tail distribution) that they mention above? These seem more important limitations to assembly, and one that I don't think sequins will address. In short, I think it's important to be clear here what spike-ins can tell you and what they can't; the language in this section is important.

We thank the reviewer for this feedback. We have provided the expanded analysis of this results as follows:

"De novo assembly of metagenome samples can be confounded by both inherent sample complexity, as well as technical factors introduced during library preparation, sequencing or analysis. Sequins can distinguish between these outcomes and thereby assist in quality control and troubleshooting. To demonstrate this, we performed de novo assembly on all samples using Ray Meta, and evaluated the quality of each assembly using MetaQUAST (supplying the sequin reference sequences). The N50 values for contigs aligned to sequins (2,272 – 2,879 nt) indicated that library preparation, sequencing and bioinformatic assembly were performed successfully (Supplementary Fig. 7c). Conversely, the N50 values for unaligned contigs were poorer and less consistent (697 – 3,659 nt), indicating that inherent sample complexity limited assembly, rather than technical factors."

Minor Comments:

1. Abstract: "developed a set internal DNA"  developed a set of internal DNA"

We have corrected the text accordingly.

2. Abstract: "accompanying microbe communities"  "accompanying microbial communities"

We have amended the text as suggested.

3. Introduction, final paragraph: I think the phrase "that represents a collection of natural microbial genomes" is confusing this early in the paper. I think adding a sentence here explaining that these are sequences *not* found in "natural microbial genomes" would help the reader grasp earlier on what you are doing. I'm not sure what the right phrase here is, but this sentence as written had me thinking the synthetic sequences were sampled directly from genomes the first time I read it.

We have amended the sentence as follows to avoid confusion:

"Here, we have developed a set of 86 synthetic DNA standards termed 'sequins' (sequencing spike-ins) that represent a synthetic microbial community."

4. Results, page 2: It's unclear what "higher order genetic features" means.

We have amended the sentence thus:

*"A random ~1-10 kb section of each genome was selected and inverted to remove homology, whilst maintaining nucleotide composition, GC content, and the distribution of repetitive and unique sequences (**Fig. 1a**). Sequences were queried against the BLAST non-redundant nucleotide collection (*nt*) database in order to ensure they had no significant homology ($E\text{-value} < 0.01$) with any known natural sequences."*

5. Fig 1d, 1e: how were dashed lines fit?

The lines in **Fig. 1(d)** and **(e)** were fit using Richard's five-parameter dose-response curve (using GraphPad® Prism). This is now included in the **Fig. 1** legend.

6. Fig 2c (and elsewhere): I suggest using "sequin assembly (%)" on the y-axis. In the legend, you use the phrase, "fraction of each sequin de novo assembled" and I think this makes it much clearer and suggest you be consistent in axis label with calling these sequins in y axis. What are error bars in this plot?

We have amended the y-axis to "Sequin assembly (%)", as suggested, and the error bars represent the standard deviation between replicates, which we have now clarified in the figure legend. In **Figure 1(c)** and **(e)**, we have left the y-axis as "Genome assembly (%)" given these plots compare the assembly of sequins with real microbial genomes.

7. Fig 3 legend: "between replicates"  among replicates

We have revised the figure legend accordingly.

8. Results, page 6: The high indel rates near homopolymers for the MinION sequencing is an interesting observation, as is the fact that indels are not symmetrically distributed across the homopolymer. Has this been observed before? If so, would be nice to point that out. If this is novel, that would also be nice to point out.

Higher indel rates in homopolymers have been observed by several groups previously, and the asymmetric error profile reflects the directionality by which the read is aligned to the genome. We have now cited these previous references in the manuscript as follows:

"We next measured sequencing error rates, finding that MinION had a mismatch error rate of 7.12% (compared to 0.127% for Illumina) and an indel rate of 8.71% (compared to 0.00770% for Illumina). Notably, MinION sequencing suffered significantly higher indel rates in homopolymeric sequences (mean indel rate = 16.7%) compared to other regions (mean = 7.69%; unpaired t-test with Welch's correction, p-value < 0.0001), with characteristic sequence coverage drops at the upstream end of homopolymer tracts (Supplementary Fig. 5c). This phenomenon has also been reported by others^{17,27,28}."

9. Results, page 7: "assembly is due to sampe complexity"  ... sample complexity

This has been corrected in the manuscript.

10. Some sequins are outliers in *de novo* assembly coverage (Fig. 1) and error rate (Fig S4). Do the authors recommend all sequins for use, or did they remove any designed sequins from what they are making available?

We initially synthesized a larger pool of sequins for this study, of which a small number failed our initial quality control testing and were omitted from manufacture. However, all 86 sequins described in the study (and which are available in the formulated mixtures) have been verified and we recommend them all for use. While there is some variation in sequencing coverage, *de novo* assembly and sequencing error rates between individual sequins, this is to be expected due to their diverse sequence characteristics, and indicates the variation that will be similarly observed in accompanying microbial samples.

References

1. Quince, C., Walker, A.W., Simpson, J.T., Loman, N.J. & Segata, N. Shotgun metagenomics, from sampling to analysis. *Nat. Biotechnol.* **35**, 833-844 (2017).
2. Thomas, T., Gilbert, J. & Meyer, F. Metagenomics - a guide from sampling to data analysis. *Microb. Inform. Exp.* **2**, 1-12 (2012).
3. Venter, J.C. *et al.* Environmental Genome Shotgun Sequencing of the Sargasso Sea. *Science* **304**, 66-74 (2004).
4. Howe, A.C. *et al.* Tackling soil diversity with the assembly of large, complex metagenomes. *Proceedings of the National Academy of Sciences* **111**, 4904-4909 (2014).
5. The Human Microbiome Project Consortium. Structure, function and diversity of the healthy human microbiome. *Nature* **486**, 207-214 (2012).
6. Gargis, A.S., Kalman, L. & Lubin, I.M. Assuring the Quality of Next-Generation Sequencing in Clinical Microbiology and Public Health Laboratories. *J. Clin. Microbiol.* **54**, 2857-2865 (2016).
7. Hug, L.A. *et al.* A new view of the tree of life. *Nature Microbiology* **1**, 16048 (2016).
8. Nayfach, S. & Pollard, K.S. Toward Accurate and Quantitative Comparative Metagenomics. *Cell* **166**, 1103-1116 (2016).
9. Gibbons, S.M. & Gilbert, J.A. Microbial diversity—exploration of natural ecosystems and microbiomes. *Current Opinion in Genetics & Development* **35**, 66-72 (2015).
10. Mason, C.E., Afshinnekoo, E., Tighe, S., Wu, S. & Levy, S. International Standards for Genomes, Transcriptomes, and Metagenomes. *J. Biomol. Tech.* **28**, 8-18 (2017).
11. Hardwick, S.A., Deveson, I.W. & Mercer, T.R. Reference standards for next-generation sequencing. *Nat. Rev. Genet.* **18**, 473-484 (2017).
12. Jones, M.B. *et al.* Library preparation methodology can influence genomic and functional predictions in human microbiome research. *Proceedings of the National Academy of Sciences* **112**, 14024-14029 (2015).
13. Angelakis, E. *et al.* Glycans affect DNA extraction and induce substantial differences in gut metagenomic studies. *Scientific Reports* **6**, 26276 (2016).
14. Risso, D., Ngai, J., Speed, T.P. & Dudoit, S. Normalization of RNA-seq data using factor analysis of control genes or samples. *Nat. Biotechnol.* **32**, 896-902 (2014).
15. Stämmeler, F. *et al.* Adjusting microbiome profiles for differences in microbial load by spike-in bacteria. *Microbiome* **4**, 28 (2016).
16. Singer, E. *et al.* High-resolution phylogenetic microbial community profiling. *ISME J* **10**, 2020-2032 (2016).
17. Loman, N.J., Quick, J. & Simpson, J.T. A complete bacterial genome assembled de novo using only nanopore sequencing data. *Nat Meth* **12**, 733-735 (2015).
18. Quick, J. *et al.* Real-time, portable genome sequencing for Ebola surveillance. *Nature* **530**, 228 (2016).
19. Goldberg, B., Sichtig, H., Geyer, C., Ledebner, N. & Weinstock, G.M. Making the Leap from Research Laboratory to Clinic: Challenges and Opportunities for Next-Generation Sequencing in Infectious Disease Diagnostics. *mBio* **6**(2015).
20. Moran-Gilad, J. *et al.* Proficiency testing for bacterial whole genome sequencing: an end-user survey of current capabilities, requirements and priorities. *BMC Infectious Diseases* **15**, 174 (2015).
21. Meyer, F. *et al.* The metagenomics RAST server – a public resource for the automatic phylogenetic and functional analysis of metagenomes. *BMC Bioinformatics* **9**, 1-8 (2008).
22. Gandolfo, L.C. & Speed, T.P. RLE plots: Visualizing unwanted variation in high dimensional data. *PLOS ONE* **13**, e0191629 (2018).

23. Benson, D.A. *et al.* GenBank. *Nucleic Acids Res* **42**, D32-7 (2014).
24. Letunic, I. & Bork, P. Interactive tree of life (iTOL) v3: an online tool for the display and annotation of phylogenetic and other trees. *Nucleic Acids Research* **44**, W242-W245 (2016).
25. Glassing, A., Dowd, S.E., Galandiuk, S., Davis, B. & Chiodini, R.J. Inherent bacterial DNA contamination of extraction and sequencing reagents may affect interpretation of microbiota in low bacterial biomass samples. *Gut Pathogens* **8**, 24 (2016).
26. Salter, S.J. *et al.* Reagent and laboratory contamination can critically impact sequence-based microbiome analyses. *BMC Biology* **12**, 87 (2014).
27. Sović, I. *et al.* Fast and sensitive mapping of nanopore sequencing reads with GraphMap. *Nature Communications* **7**, 11307 (2016).
28. Jain, M. *et al.* Improved data analysis for the MinION nanopore sequencer. *Nat Meth* **12**, 351-356 (2015).
29. Sohn, M.B., Du, R. & An, L. A robust approach for identifying differentially abundant features in metagenomic samples. *Bioinformatics* **31**, 2269-2275 (2015).
30. David, L.A. *et al.* Host lifestyle affects human microbiota on daily timescales. *Genome Biology* **15**, R89 (2014).

REVIEWERS' COMMENTS:

Reviewer #1 (Remarks to the Author):

In the revised version of their manuscript, Hardwick et al thoroughly addressed the reviewer's comments. I'm looking forward to seeing this study published and testing the sequins.

A few minor pending comments:

- Supplemental Material: please ensure that the supplemental material is legible, such as table headers (currently cut off in the pdf) and the resolution of Supplementary Figures and their labels is high (currently not the case, at least not in the documents I downloaded)
- "As well as enabling quality control and inter-sample normalization, reference standards are also essential for the development and optimization of new sequencing technologies^{11,14}." Replace with "In addition to enabling quality control and inter-sample normalization, reference standards are essential for the development and optimization of new sequencing technologies^{11,14}."
- "While samples still did not cluster perfectly by type, it must be borne in mind that the replicates were collected from three separate natural and regenerated sites." – did you mean "kept" in mind?
- "Nonetheless, the fact that samples clustered loosely by type but seemingly not at all by environment is an interesting biological observation, and implies that regeneration of these saltmarsh sites after initial development did not have a discernible effect on the microbial ecology of the samples." Remove "seemingly" – they either do or they don't.

Reviewer #2 (Remarks to the Author):

All of my concerns have been addressed with one exception under which I group a few more comments and questions. These all surround Figure 3 and associated text.

- 1) Please add somewhere in the legend (or discussion of this figure in the results) that this is based on phylum-level abundance estimates from read mapping.
- 2) The drops in coefficient of variation are nice. However, for the new panel 3b, can you clarify what is calculated, here / how this is calculated? I'm being dense; Is this for phylum-level relative abundances coefficient of variation, calculated for each phylum within each sample type? If I'm understanding this, then a distribution of the data or the data itself should probably be plotted, instead of just mean values.
- 3) My concerns about using RLE were related to the fact 1) this might not be considered high-dimensional data, and 2) that it might not be a safe assumption that *most* phyla will be at similar levels across the various sample types, as compared to gene expression studies with thousands of genes and more controlled manipulations. That is, moving from the soil to the rhizosphere might be the equivalent of the antibiotic exposure you cite. Can you simply state, even in the supplemental figure S8, how many phyla are being considered here via the mapping to MG-RAST? In figure S9, we are talking just 3 domains, correct? Really, this is that rambling reviewer comment that doesn't ask you to do anything of substance. I'm just wondering if the normalization is overly aggressive in these cases, as the relationship in the PCA do change significantly in some cases and there is no way of knowing, as you point out, which form of sample beta-diversity clustering is "correct."

"to compare variation between replicates"  "among replicates" (there are 3 in each category, no?)

4) Lastly, the added sentence to the Results section echoes almost word for word from citation 22. Though I'm sure inadvertent, this seems inappropriate.

Added sentence:

"RLE plots, while originally devised for gene expression data, can also be used to reveal unwanted variation in many other kinds of high-dimensional data, and are particularly useful for assessing whether a normalization procedure has been successful [22]."

Ref 22:

Introduction:

"RLE plots, while originally devised for microarray data, can also be used to reveal unwanted variation in many other kinds of high dimensional data"

Conclusion:

"RLE plots are particularly useful for assessing whether a normalization procedure has been successful"